# Interplay of entanglement structures and stabilizer entropy in spin models

**M. Viscardi[1,3, ⋆], M. Dalmonte[2], A. Hamma[1,3,4], E. Tirrito[2, †]**

**1** Dipartimento di Fisica "E. Pancini", Università di Napoli Federico II, Monte S. Angelo, 80126, Napoli, Italy
**2** The Abdus Salam International Center for Theoretical Physics (ICTP), Strada Costiera 11, 34151, Trieste, Italy
**3** INFN, Sezione di Napoli
**4** Scuola Superiore Meridionale, Largo S. Marcellino 10, 80138, Napoli, Italy

⋆ michele.viscardi@unina.it , † etirrito@ictp.it

## Abstract

**Understanding the interplay between nonstabilizerness and entanglement is crucial for uncovering the fundamental origins of quantum complexity. Recent studies have proposed entanglement spectral quantities, such as antiflatness of the entanglement spectrum and entanglement capacity, as effective complexity measures, establishing direct connections to stabilizer Rényi entropies. In this work, we systematically investigate quantum complexity across a diverse range of spin models, analyzing how entanglement structure and nonstabilizerness serve as distinctive signatures of quantum phases. By studying entanglement spectra and stabilizer entropy measures, we demonstrate that these quantities consistently differentiate between distinct phases of matter. Specifically, we provide a detailed analysis of spin chains including the XXZ model, the transverse-field XY model, its extension with Dzyaloshinskii-Moriya interactions, as well as the Cluster Ising and Cluster XY models. Our findings reveal that entanglement spectral properties and magic-based measures serve as intertwined, robust indicators of quantum phase transitions, highlighting their significance in characterizing quantum complexity in many-body systems.**

# 1   Introduction

Quantum spin models, in particular spin chains, have long served as paradigmatic models in the study of Quantum Many-Body (QMB) systems, bridging key concepts in condensed matter physics [1–3] and quantum information [4–6]. Their simple formulation, coupled with their ability to span a broad range of physical behaviors, makes them precious tools for probing the properties of complex quantum systems through both analytical and numerical approaches [7,8].

A major factor behind the enduring importance of spin chains is their strong connection to experimental realizations. Hamiltonians such as the transverse-field Ising model [7], the XXZ model [9], and Kitaev chains [10], just to mention a few important examples, have been successfully implemented in various platforms [11], including Rydberg atom arrays [12], trapped ions [13], and superconducting qubits [14]. These systems offer unique opportunities to test theoretical predictions and reveal novel quantum phenomena. The rapid advancements in these experimental platforms, particularly in the context of quantum computation, have further fueled the interest in spin models, especially in terms of the computational quantum resources of entanglement and nonstabilizerness (often referred to as *magic*).

These two resources have been, due to their importance, extensively analyzed within the framework of Quantum Resource Theories (QRTs) [15]. The resource theory of entanglement [16–20] has found extensive applications in quantum communication [21]. It has been instrumental in addressing fundamental questions, on the distillability of entanglement [22,23] and the distinguishability of quantum states [24]. In QMB systems, entanglement has been

widely investigated because of its connections with classical simulability [25, 26] and the description of novel quantum phases of matter [27–30]. It has also been extensively studied in the context of quantum phase transitions (QPTs), where it has emerged as a powerful tool for understanding the non-classical correlations and universal properties of QMB systems. In particular, entanglement, often quantified via the entanglement entropy, has been proven to be sensitive to critical points across a variety of models [4], exhibiting scaling laws that are governed by the central charge of the underlying Conformal Field Theory (CFT) of the model. While a comprehensive review of entanglement and its connections to CFTs in critical systems lies beyond the scope of this work, we refer interested readers to [5, 31] for detailed discussions.

It is important to notice, however, that conventional measures of entanglement, such as the entanglement entropy or Rényi entropies, often fall short in fully capturing the complexity of quantum systems. In fact, even though entanglement is a source of computational complexity in many numerical methods (such as for Matrix Product States (MPS) and other tensor network methods), it has been proven to not being sufficient for achieving quantum advantage [32]. Indeed, the stabilizer formalism can be exploited to efficiently simulate even maximally entangled states [33, 34]. What prevents the stabilizer formalism from being efficient for any quantum state is, of course, nonstabilizerness [35].

The importance of nonstabilizerness in quantum computation motivated the development of magic resource theories [36], which have since been applied in many different settings. One of the most important examples is that of magic state distillation, where magic RTs have been adopted to prove bounds on distillability [37–40] and on direct fidelity and purity estimation [41, 42]. More recently, magic RTs have also found application in the theory of learning for quantum states, where it was adopted in the design of efficient learning algorithms for some classes of states and for proving bounds on the tomographic sampling/measurement complexity and learning efficiency [43–45]. Magic RTs have also been successfully applied to study quantum complexity in many other different fields, including: relativistic quantum information [46], particle physics (in particular, in systems of dense neutrinos [47] and systems of top quarks [48]), nuclear physics (such as in $p$-shell and $sd$-shell nuclei and in nuclear and hypernuclear forces [49, 50]), and quantum gravity [51–53]. Moreover, a notable progress has been made in experimental measurements of magic resources [54–57]. Building on the profound insights gained from entanglement, recent studies have also been analyzing various measures of nonstabilizerness in QMB systems, especially in spin models, examining whether the latter is connected to quantum phase transitions. Here, a significant portion of the magic RTs' literature has focused on developing powerful and efficient techniques for computing magic monotones, allowing for quantifying nonstabilizerness for large system sizes [58–71]. In parallel with numerical efforts, we also note that the relevance of magic to quantum criticality has also recently been discussed in the context of CFT [72]. In the context of quantum thermodynamics, quantum spin models - e.g., SYK models - have been studied for their potential to show quantum advantage as a quantum battery [73–77], with a super extensive charging power [78]. The quantum advantage of quantum spin batteries could be related to an optimal usage of both entanglement [79] and non stabilizer resources [80–82].

In recent years, a clearer picture of quantum complexity has emerged: the absence of either entanglement or magic forbids attaining quantum advantage in quantum computation, allowing for an efficient classical simulation of such quantum states. Furthermore, it has been proven in [83] that the Hilbert space of a quantum system can be operationally divided into two distinct phases: one where entanglement estimation/manipulation tasks are efficient (the entanglement dominated phase) and one where these tasks are intrinsically hard (the magic dominated phase). In other words, the predominant presence of nonstabilizerness in a quantum state induces a complex entanglement structure, impinging entanglement manipulation

tasks and its efficient estimation. The resulting picture, then, is that quantum complexity must originate from the intricate interplay of entanglement and nonstabilizerness [51,83–86]. This observation provided further motivations for the analysis of entanglement spectral quantities. In particular, Entanglement Spectrum Statistics (ESS) [84] have been linked to universality in quantum computations and have also been shown to be deeply connected to many interesting phenomena in QMB systems, such as: the Eigenvalue Thermalization Hypothesis (ETH) [87–91], the Anderson localization and Many Body Localization (MBL) [92,93].

Considering the importance of spin models in quantum computation, in this work we aim at providing a solid characterization of their ground states. Our work complements, both in spirit and in terms of factual results, recent computational efforts in investigating magic in many-body systems, as we detail below. In particular, we present a combined investigation of properties related to moments of the entanglement spectrum, and magic: this is important to ascertain not only the connection between any of the two and physical phenomena (e.g., critical behavior), but, most importantly, how strongly those are connected. In order to illustrate such connection between entanglement and magic at a simple level, we have included a review of magic in spin systems below. There, we cover two-spin models, where the above mentioned relation can be addressed analytically.

The structure of the paper is as follows. Section 2 opens with an overview of magic resource theories, highlighting their relationship with entanglement. Here, both the antiflatness of the entanglement spectrum and the moments of the entanglement Hamiltonian are introduced and discussed in detail. In Section 3, we present an in-depth review of nonstabilizerness as a tool for probing criticality in QMB systems, particularly in spin systems, systematically organizing the literature according to the selected magic monotones. The main contributions of this work are reported in Section 4, where we present the complete phase diagrams of various spin models, both with and without cluster interactions, with respect to selected measures of entanglement, nonstabilizerness and, most importantly, quantum complexity. The study includes models with nearest-neighbor interactions, such as the XXZ model, the XY model, both with and without Dzyaloshinskii-Moriya interactions, as well as models with cluster interactions, such as the cluster Ising [94–96] and cluster XY models [97]. We find that entanglement spectral quantities, like the antiflatness and the capacity of entanglement, are consistently sensitive to the critical points of the latter mentioned models, proving to be useful diagnostic tools to understand quantum complexity in QMB systems, even at finite sizes.

## 2   Entanglement, Magic, and their connection

One of the main goals in quantum information is to find a thorough picture of quantum complexity, which is complementary to answering the question of where does quantum computational advantage come from. QRTs provide us with a precise mathematical framework to quantitatively study properties of quantum systems under operational constraints [15,98]. For this reason, QRTs have become a cornerstone for uncovering the fundamental aspects of quantum technologies. For example, the resource theory of entanglement, where Local Operations and Classical Communication (LOCC) are considered as free operations, has been extensively applied in practical settings such as quantum communication and cryptography, providing, among other results, bounds on the efficiency of entanglement distillation protocols [99]. Recently, [100], the connection between entanglement and magic has been systematically studied in the context of random states in the Hilbert space, showing the rich interplay between the two, in spite of the lack of statistical correlations. This suggests that correlations between entanglement and magic would be specific of structured states like the ground states of local Hamiltonians and the physics of the specific model.

Another well-studied resource theory is that of nonstabilizerness, which has been vastly used in the context of quantum computation, where stabilizer error-correcting codes play a fundamental role in achieving fault-tolerant computations. In this section, we provide a brief overview of the resource theory of magic, introducing foundational concepts such as the Pauli and Clifford groups, as well as the notion of magic monotones. We also explore the relationship between entanglement and nonstabilizerness in Section 2.2 and Section 2.3, where we examine Rényi entropy capacities and the antiflatness of the entanglement spectrum.

## 2.1 Resource Theory of Magic

As previously mentioned, one of the most promising paradigms of quantum computation is that of fault-tolerant quantum computation [33], which aims to reduce the impact of random noise by implementing error-correcting schemes. The latter often relies on the stabilizer formalism, which will now be summarized.

Let us start by considering a $N$ qubit system with Hilbert space $\mathcal{H} = \otimes_{j=1}^{N} \mathcal{H}_j$. The $N$-qubit Pauli group $\mathcal{P}_N$ encompasses all the possible Pauli strings with overall phases $\pm 1$ and $\pm i$. More precisely, we can write:

$$\mathcal{P}_N = \left\{ e^{i\theta \frac{\pi}{2}} \sigma_{j_1} \otimes \cdots \otimes \sigma_{j_k} \otimes \cdots \otimes \sigma_{j_N} \,|\, \theta, j_k = 0, 1, 2, 3 \text{ and } k = 1, 2, \ldots, N \right\}. \tag{1}$$

A pure $N$-qubit state is called a stabilizer state if it satisfies the following conditions. Specifically, a pure stabilizer state is associated with an Abelian subgroup $\mathcal{S} \subset \mathcal{P}_N$ containing $2^N$ elements such that $S|\psi\rangle = |\psi\rangle$ for all $S \in \mathcal{S}$ (i.e. $S$ stabilizes $|\psi\rangle$). Alternatively, we can define a stabilizer state using Clifford unitaries. The Clifford group $\mathcal{C}_N$ is defined as the normalizer of the $N$-qubit Pauli group:

$$\mathcal{C}_N = \left\{ U \,|\, UPU^\dagger \in \mathcal{P}_N, \forall\, P \in \mathcal{P}_N \right\}. \tag{2}$$

The Clifford group is generated by combining the Hadamard gate, the $S$ gate (a $\pi/2$ phase gate), and the CNOT gate. Stabilizer states encompass all the states that can be generated by Clifford operations acting on the computational basis state $|0\rangle^{\otimes N}$.

Despite their very rich structures and large (in fact, possibly maximal) entanglement, it is well known that stabilizer states are not sufficient for quantum advantage. Indeed, the Gottesman-Knill theorem [34, 101, 102] states that any quantum computation using only stabilizer states and Clifford unitaries can be efficiently simulated by a classical computer. In other words, nonstabilizerness is essential for achieving both quantum computational universality and advantage. As an example, extending the stabilizer gate set with a non-stabilizer gate enables quantum circuits to perform universal quantum computation. For instance, adding the $T$ gate:

$$T = \begin{pmatrix} 1 & 0 \\ 0 & e^{i\pi/4} \end{pmatrix}, \tag{3}$$

is sufficient to reach universality [39]. Unfortunately, it has been proven that a fault-tolerant scheme with a set of gates that are both universal and transversal does not exist [103].

A workaround to this issue is to supply magic states [39], such as:

$$|T\rangle\langle T| = \frac{1}{2}\left( I + \frac{X + Z}{\sqrt{2}} \right), \tag{4}$$

$$|H\rangle\langle H| = \frac{1}{2}\left( I + \frac{X + Y}{\sqrt{2}} \right), \tag{5}$$

$$|F\rangle\langle F| = \frac{1}{2}\left( I + \frac{X + Y + Z}{\sqrt{3}} \right), \tag{6}$$

and to act on them with stabilizer operations. Indeed, given a single copy of the Hadamard eigenstate $|H\rangle$ or the Clifford equivalent $T$-state $|T\rangle$, we can perform a deterministic $T$ gate using state injection. Therefore, full universality can be achieved given stabilizer operations and an appropriate supply of magic states.

In this context, quantifying the non-Clifford resources required to prepare a given state is crucial. The resource theory of nonstabilizerness identifies stabilizer states as the free set, with the amount of magic in a state measured through magic monotones. The properties required for a good monotone $\mathcal{M}$ of nonstabilizerness are:

1. faithfulness to stabilizer states: $\mathcal{M}(|\psi\rangle) = 0$ iff $|\psi\rangle$ is a stabilizer state

2. non-increasingness under stabilizer operations $\mathcal{E}$[1]: $\mathcal{M}(\mathcal{E}(|\psi\rangle)) < \mathcal{M}(|\psi\rangle)$

3. sub-additivity in the tensor product: $\mathcal{M}(|\psi\rangle \otimes |\phi\rangle) \leq \mathcal{M}(|\psi\rangle) + \mathcal{M}(|\phi\rangle)$.

## 2.2 Brief review on Rényi entropies capacity and moments

As mentioned in Sec. 1, the origin of quantum complexity is found in the interplay of both entanglement and nonstabilizerness. Recently, these two quantities were put in contact via many entanglement spectral quantities. In this section, we introduce such complexity measures, which will underpin our analysis throughout the paper.

Consider a quantum system in a state $\rho$ defined on a Hilbert space $\mathcal{H} = \mathcal{H}_A \otimes \mathcal{H}_{\bar{A}}$, where $A$ and $\bar{A}$ denote two complementary subsystems. To obtain the reduced density matrix of subsystem $A$, we trace out the degrees of freedom associated with subsystem $\bar{A}$. The reduced density matrix $\rho_A$ is given by:

$$\rho_A = \mathrm{Tr}_{\bar{A}}(\rho),$$

where $\mathrm{Tr}_{\bar{A}}$ denotes the partial trace over the Hilbert space $\mathcal{H}_{\bar{A}}$.

From this, we can compute the $n$-Rényi entropy as:

$$S_n(\rho_A) = \frac{1}{1-n} \log\left(\mathrm{Tr}\,\rho_A^n\right), \tag{7}$$

for integer $n > 1$. One can analytically continue for non-integer $n$. The most important limit is to $n \to 1$:

$$S(\rho_A) = \lim_{n \to 1} S_n = \lim_{n \to 1} \frac{1}{1-n} \log\left(\mathrm{Tr}\,\rho_A^n\right) = -Tr\left(\rho_A \log \rho_A\right), \tag{8}$$

where the Rényi entropy becomes the entanglement entropy. We can also express the entanglement entropy as the expectation value of the entanglement Hamiltonian $H_A = -\log \rho_A$ with respect to $\rho_A$:

$$\langle H_A \rangle_{\rho_A} = \mathrm{Tr}(\rho_A H_A) = S(\rho_A). \tag{9}$$

Eq. 9 naturally suggests to also study higher moments of $H_A$. We can, for instance, compute the second moment (i.e. the variance) of $H_A$, which is:

$$Var(H_A) = \langle H_A^2 \rangle - \langle H_A \rangle^2 = \mathrm{Tr}\left(\rho_A \log^2 \rho_A\right) - \mathrm{Tr}^2\left(\rho_A \log \rho_A\right) = C_E. \tag{10}$$

In analogy with its thermodynamical cousin, this quantity is known in literature as entanglement capacity and has recently gained some interest [52, 104–107]. Essentially, it can be thought of as a measure of how non-flat the entanglement spectrum of a state is. Indeed, $C_E = 0$ if and only if the associated entanglement spectrum is flat, i.e. $\lambda_\alpha = 1/\chi$ for some

---

[1]It is important to notice that free operations may change in dependence of the considered magic monotone. The most studied resource thoeries of nonstabilizernes have as their free sets the set of pure stabilizer states. For such resource theories, Clifford unitaries are always nonincreasing, and therefore free, operations, independently of the particular monotone.

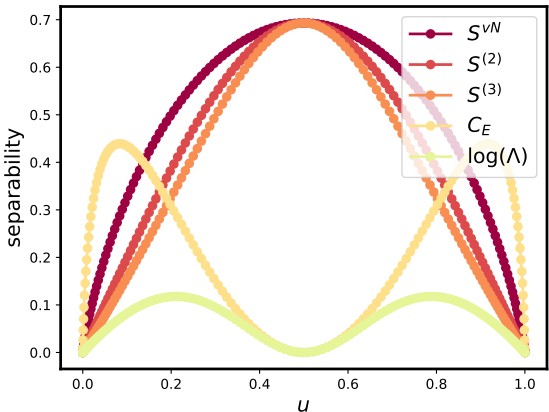

Figure 1: **Rényi entropies, entanglement capacity and log-ratio of moments of the reduced state:** We present the variation of the entanglement capacity, Rényi entropies, and the log-ratio ($\log \Lambda$) of the moments of the single-qubit reduced state derived from the two-qubit state in Eq. 15, as a function of $u$. Notably, the behavior of $C_E$ [106, 107] and $\log \Lambda$ differs significantly from that of the Rényi entropies. While the Rényi entropies attain their maximum value at $u = 0.5$ (corresponding to the maximally entangled state), where both the capacity and $\log \Lambda$ vanish, the latter two exhibit peaks at intermediate values of $u$, corresponding to partially entangled states.

integer $1 \leq \chi \leq \min(2^{|A|}, 2^{|\bar{A}|})$, with $\lambda_\alpha$ being the Schmidt coefficients of the reduced state. The capacity of entanglement is also related to Rényi entropies by the following relation:

$$C_E = \lim_{n \longrightarrow 1} n^2 \partial_n^2 ((1 - n) S_A^{(n)}). \tag{11}$$

Other closely related quantities capturing information on the entanglement can be obtained by considering the moments $p_k = \text{Tr}\left[(\rho_A)^k\right]$ of the density matrix. In this work, we will consider the ratio and the difference of the third moment and the square of the second moment. In formulas:

$$\mathcal{F}_A = \text{Tr}\left[\rho_A^3\right] - \text{Tr}^2[\rho_A^2]. \tag{12}$$

$$\Lambda_A = \frac{\text{Tr}\left[\rho_A^3\right]}{(\text{Tr}\left[\rho_A^2\right])^2}, \tag{13}$$

and:

$$\log \Lambda_A = \log \text{Tr}\left[\rho_A^3\right] - 2 \log\left(\text{Tr}\left[\rho_A^2\right]\right). \tag{14}$$

$\Lambda_A$, $\log \Lambda_A$ and $\mathcal{F}_A$ are all valid measures of the flatness of the entanglement spectrum. Specifically, for states with a flat spectrum, $\Lambda_A = 1$ ($\log \Lambda_A = 0$) and $\mathcal{F}_A = 0$, while for states with non-flat spectra, $\Lambda_A > 1$ ($\log \Lambda_A > 0$) and $\mathcal{F}_A > 0$. The antiflatness is closely related to nonstabilizerness, a connection we will further explore in Section 2.3.

As an example, let us consider a two-qubits system ($N = 2$) and the simple state:

$$|\psi\rangle = \cos(\beta/2)|01\rangle + e^{i\chi} \sin(\beta/2)|10\rangle \tag{15}$$

with $\beta \in [0, \pi]$, $\chi \in [0, 2\pi]$ . If one traces out the second qubit, the reduced density matrix becomes $\rho = \text{diag}(u, 1-u)$, where $u = \cos^2(\beta/2)$. A direct computation gives:

$$S(\rho) = -(1 - u)\log(1 - u) - u \log u \tag{16}$$

$$C_E(\rho) = (1 - u)\log^2(1 - u) + u \log^2(u) - [(1 - u)\log(1 - u) + u \log u]^2 \tag{17}$$

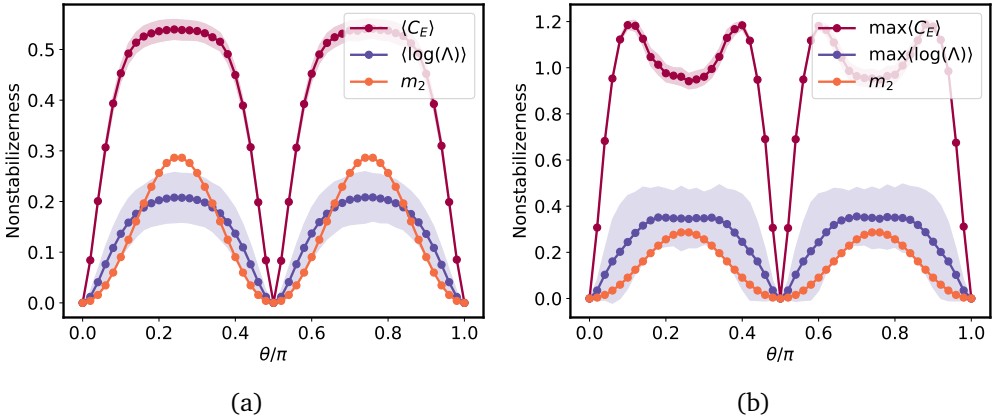

Figure 2: **Antiflatness, capacity of entanglement and stabilizer Rényi entropy:**
(2a) The stabilizer Rényi entropy (SRE) as a function of the parameter $\theta$ for the
initial state defined in Eq. 21. The behaviour of SRE is periodic in $\theta$ and it reaches
a maximum for $\theta = \pi/4$. Moreover, we plot the average over different realizations
and time of $\Lambda$ and the average of the $C_E$ as a function of parameter $\theta$. (2b) We plot
the average over the trajectories and time of the maximum value of $\log\langle\Lambda\rangle$ and of $C_E$
as a function of parameter $\theta$.

$$S^{(n)}(\rho) = (1-n)^{-1}\log\left((1-u)^m + u^m\right) \tag{18}$$

$$\Lambda(\rho) = \left(u^3 + (1-u)^3\right)\left(u^2 + (1-u)^2\right)^{-2} \tag{19}$$

The variation of the entanglement entropy, the capacity of entanglement, the $\log\Lambda$ and the
$m = 2, 3$-Rényi entropies with respect to $u$ is shown in Fig. 1. At $u = 0$ and $u = 1$, the state in
Eq. (15) is factorized, which implies that all the entanglement measures will be vanishing. At
$u = 0.5$, the state becomes maximally entangled (EPR state), with the entanglement entropy
and all Rényi entropies achieving a value of $\log 2$. In contrast, both the entanglement capacity
and $\log\Lambda$ vanish. This behavior is expected, as the reduced state at $u = 0.5$ is the completely
mixed state (which has a flat spectrum), and the system, already maximally entangled, cannot
gain additional entanglement through unitary operations. On the same line, the entanglement
capacity peaks up a value $C_{max} = 0.4392$ at $u = 0.0832$ and $u = 0.9168$ which is in fact a
partially entangled state.

## 2.3 Magic and antiflatness

The antiflatness quantifies deviations of the entanglement spectrum from being flat across a
given bipartition. Operationally, it captures how far the classical probability distribution ob-
tained by measuring the subsystem in its Schmidt basis departs from the uniform distribution.
A vanishing antiflatness therefore guarantees that the measurement statistics in the Schmidt
basis are perfectly uniform, but this property is strictly basis-dependent: it does not imply
that the same state yields a flat (or even approximately flat) probability distribution when
measured in any other basis.

    As illustrated in Fig. 1, the entanglement spectrum of a bipartite pure state is flat whenever
the state is either fully factorized or maximally entangled. Away from these two limiting
cases, deviations from flatness arise precisely due to nonstabilizerness. Recent works have
established a direct and quantitative connection between antiflatness and magic. In particular,
antiflatness links magic—an intrinsic property of the full state—to bipartite entanglement and,
consequently, to the spectrum of the reduced density matrix. Ref. [108] showed that for any

pure state $|\psi\rangle$, the antiflatness averaged over its Clifford orbit is proportional to the stabilizer linear entropy[2]:

$$\langle \mathcal{F}_A(\Gamma |\psi\rangle)\rangle_{C_n} = c(d, d_A) M_{\text{lin}}(|\psi\rangle), \tag{20}$$

where $d$ is the Hilbert-space dimension of the full system and $d_A$ that of subsystem $A$. This relation holds for any bipartition, a fact encoded in the bipartition-dependent constant $c(d, d_A)$. A direct consequence is that pure stabilizer states—whose stabilizer linear entropy vanishes— exhibit a flat entanglement spectrum throughout their entire Clifford orbit. Antiflatness is therefore invariant under Clifford operations. More broadly, this theorem establishes a rigorous bridge between entanglement and magic, with implications for tensor-network descriptions [70, 109] and for participation-ratio based diagnostics [110]. In particular: (i) without entanglement, the reduced density operator cannot display antiflatness, and (ii) for highly entangled states— typical in many-body Hilbert spaces—antiflatness can be used as an efficient estimator of $M_{\text{lin}}$, i.e., of the underlying magic content.

In Fig. 2a we numerically investigate the dependence of the SRE as a function of a parameter $\theta$ in the initial product state:

$$|\psi\rangle = \bigotimes_{i=1}^{N} \left( \frac{|0\rangle + e^{i\theta}|1\rangle}{\sqrt{2}} \right), \tag{21}$$

We start from the state in Eq. 21 with $N = 20$ qubits and compute its 2-Stabilizer Rényi Entropy (2-SRE) [111], a magic monotone for pure states. We label by $m_2$ the 2-SRE density, defined just as the 2-SRE of a state divided by the number of qubits in the system. We see that for $\theta = 0, \pi/2$ and $\pi$ the 2-SRE is vanishing, since for this choices of $\theta$ the initial states are stabilizers. Conversely, for $\theta = \pi/4, 3\pi/4$, the 2-SRE density reaches the maximum for product states, which is simply the value of the 2-SRE for a single $T$-state. In Fig. 2 we also numerically study the dependence of $\Lambda$, the capacity $C_E$ on the initial states defined in Eq. 21 with respect to random Clifford operations. The states in Eq. 21 are product states, and because of this they all share the following initial values: $\Lambda = 1$ and $C_E = 0$. We, then, apply the protocol defined in [108] with $N_{\text{Iter}} = 2000$ different Clifford layers. Moreover, we performed $N_R = 1000$ different realizations and we calculated the average of $\Lambda$. In Fig. 2b we plot the the average of the maximum of $\Lambda, C_E$ and $m_2$ along the circuit and on different realizations. As we can see, the numerical data are in very good accordance with the theoretical prediction for the target states, showing the fitness of our protocol in detecting magic state.

# 3 Magic in quantum many-body systems

Nonstabilizerness has emerged as a valuable tool for studying quantum critical systems, with numerous works investigating its role in this context. The rapid growth of this field motivates a first review of the key findings in the literature. In this section, we examine nonstabilizerness in many-body spin systems, emphasizing whether the adopted measures are sensitive to the phase transitions of the models.

The first magic monotone to be studied in the context of many-body quantum systems was the *Robustness of Magic (RoM)*, of which we now briefly recall the definition and its main properties.

---

[2]An explicit expression for the stabilizer linear entropy is given in Sec. 3.3, where it is also related to the 2-Stabilizer Rényi Entropy.

### 3.1 Robustness of Magic

One of the first resource theories of magic states was developed recasting the definition of *robustness* of a state [112] in a theory that has nonstabilizerness as its resource of interest.

In this context, the set of free states is considered to be the convex hull of the set $S$ of pure stabilizer states (in symbols, conv$\{S\}$). These states are often referred to as *mixed stabilizer states*. Given a n-qubits system with Hilbert space $\mathcal{H}$, the *Robustness of Magic (RoM)* of a state $\rho$ is defined as:

$$\mathcal{R}(\rho) := \min\left\{\|x\|_1 : \rho = \sum_{i=1}^{|S|} x_i |s_i\rangle\langle s_i|, |s_i\rangle \in S\right\} \tag{22}$$

The RoM was defined for the first time in Ref. [113], and there it was also shown to be a magic monotone satisfying the following properties:

1. Faithfulness for mixed stabilizer states: $\mathcal{R}(\rho) = 1 \iff \rho \in$ conv$\{S\}$, otherwise $\mathcal{R}(\rho) > 1$

2. Sub-multiplicativity in the tensor product: $\mathcal{R}(\rho \otimes \sigma) \leq \mathcal{R}(\rho)\mathcal{R}(\sigma)$

3. Monotonicity with respect to stabilizer operations $\mathcal{E}$: $\mathcal{R}(\mathcal{E}(\rho)) \leq \mathcal{R}(\rho)$

4. Convexity: $\mathcal{R}((1-p)\rho + p\sigma) \leq (1-p)\mathcal{R}(\rho) + p\mathcal{R}(\sigma)$ with $p \in [0,1]$

#### 3.1.1 RoM in spin models

The first model to be studied through the lenses of RoM was the quantum XY model. In particular, in Ref. [114] the authors study the single- and two-spins RoM across various temperature regimes. At zero temperature regime, the authors showed the existence of a *Magic Pseudo-critical Point (MPP)*, where magic reaches its maximum value. In both the single- and two-spins cases, with any given anisotropy parameter $\gamma$, the MPP coincides with the factorization point of the model. At finite temperatures, similarly, the authors have shown that symmetry-unbroken ground states also display magic between distant qubits, although thermal fluctuations gradually reduce it.

Due to the impractical minimization procedure involved in the definition of the RoM, the literature shifted towards more numerically accessible magic measures, such as *Mana* and *Stabilizer Entropies*.

### 3.2 Mana

A useful magic monotone for composition of odd-dimensional Hilbert spaces is *Mana* [36]. Before recalling the definition of Mana, let us introduce a few preliminary notions. For a *d-level* system with Hilbert space $\mathcal{H}^d$, with $d$ being both odd and prime for simplicity, the generalized Pauli operators are described by the following expression:

$$T_{aa'} = \omega^{2^{-1}aa'} Z^a X^{a'} \tag{23}$$

where $\omega$ is the $d-$th root of the unity, $Z$ and $X$ are, respectively, the clock and the shift operators, and $2^{-1}$ is the multiplicative inverse of $2 \mod d$.

In this setting, we can write generalized Pauli strings over $\otimes_{i=1}^N \mathcal{H}_i^d$ as the tensor product of generalized Pauli operators:

$$T_a = T_{a_1 a_1'} T_{a_2 a_2'} \cdots T_{a_N a_N'} \tag{24}$$

One more preliminary notion is that of *phase space point operators* and discrete Wigner function, both of which will be exploited in the definition of Mana. The phase space point operators are operators defined as:

$$A_b = d^{-N} T_b \Big[ \sum_a T_a \Big] T_b^\dagger \tag{25}$$

It can be proven that these operators provide an Hermitian and Frobenius-norm orthogonal basis for the space of operators in $\mathcal{B}(\otimes_{i=1}^N \mathcal{H}_i^d)$. The expectation values of the phase space point operators with respect to a state $\rho$ are real coefficients, and form the so called *discrete Wigner function* of $\rho$. Indeed, any state $\rho$ can be written with respect to the phase space point operators as:

$$\rho = \sum_u W_\rho(u) A_u \tag{26}$$

By means of Hudson's theorem [115], the discrete Wigner function of a pure state $|\psi\rangle$ on a Hilbert space of odd dimension is non-negative if and only if $|\psi\rangle$ is a pure stabilizer state.

With respect to the previous definitions, the magic monotone of Mana can be defined (for both pure and mixed states) as:

$$\mathcal{M}(\rho) = \log \sum_u |W_\rho(u)| \tag{27}$$

It can be proven that the following properties hold for Mana:

- Linearity in the tensor product: $\mathcal{M}(\rho \otimes \psi) = \mathcal{M}(\rho) + \mathcal{M}(\psi)$

- Mana $\mathcal{M}(\rho)$ is upper-bounded by $\frac{1}{2}(N \log d - S_2)$, with $S_2$ being the 2-Rényi entropy of $\rho$.

### 3.2.1 Mana Entropies

The magic monotone of Mana has been redefined and studied in its Rényi generalization in Ref. [116]. Building on the definitions introduced in Sec. 3.2, *Mana Entropies* (MEs) are defined for pure states in a manner closely resembling the construction of *Stabilizer Entropies* (SEs), as:

$$\mathcal{M}_n(|\psi\rangle) = \frac{1}{1-n} \log \sum_u \frac{\Pi_\psi(u)^n}{d^N} \tag{28}$$

for $|\psi\rangle \in \mathcal{H} \simeq \mathbb{C}^{d^N}$ and $\Pi_\psi(u) = d^N W_\psi(u)^2$ being probabilities. MEs share many useful properties with SEs, such as: faithfulness to pure stabilizer states, invariance under Clifford unitaries and additivity with respect to tensor product. Also, many numerical methods, such as those grounded on MPS replica trick [59], can be easily modified for computing MEs of integer indices.

### 3.2.2 Mana and Mana Entropies in spin models

In Ref. [68] the authors investigate Mana for the ground state of the (1D) 3-state Potts model, showing that it possesses a significant amount of mana at the model's critical point. Not only, approaching criticality, the GS's mana density increases with respect to the subsystem size, suggesting that it is rooted in the subsystems' correlations. In the same paper, the authors introduce the notion of *connected component of mana density*, which is a way to estimate the non-locality of mana of a quantum state. They study this quantity in the GS of the 3-state Potts model by choosing two disconnected subsystems at a distance $\delta_x$ and increasing this. They find that far before criticality it rapidly vanishes for $\delta_x$ increasing, meaning that mana is well

366 localized. Instead, when the model is way after its critical point, the connected component
367 of mana rapidly reaches a plateau, suggesting that mana is very non-local in this phase of the
368 model.

369 Mana has also been studied in Ref. [116], where it is studied both in the setting of the
370 3-state Potts model on a periodic chain and in a non-integrable extension of it. The second
371 work confirms the mana density to be peaking at the phase transition of the model. Moreover,
372 numerical evidence shows that mutual mana exhibits universal logarithmic scaling at criticality,
373 both in the Potts model and in its non-integrable extension.

## 3.3 Stabilizer Rényi Entropy

375 The most studied magic monotone is the *Stabilizer Rényi Entropy* (SRE, or SE), which was
376 defined for the first time in Ref. [111]. This quantity does not require a minimization procedure
377 in order to be computed, and therefore it has been extensively adopted for both numerical and
378 analytical studies on the role of nonstabilizerness in QMB systems.

### 3.3.1 Definition and properties

380 The SREs are defined as the Rényi entropies of the probability distribution of a state in the Pauli
381 basis. More precisely, by labeling with $P$ a generic composition of Pauli operators (namely, a
382 Pauli string), the normalized expectation value $\Xi_P(|\psi\rangle)$ is defined as:

$$\Xi_P(|\psi\rangle) := d^{-1} \langle\psi| P |\psi\rangle^2 \tag{29}$$

383 where $d$, here, is the dimension of the total Hilbert space (e.g. for a $n$-qubits system $d = 2^n$).
384 This can be interpreted as the probability of finding $P$ in the representation of a state $|\psi\rangle$.
385 Hence $\{\Xi_P(|\psi\rangle)\}$ is a well-defined probability distribution. It is now possible to take the
386 $\alpha-$Rényi entropy of $\{\Xi_P(|\psi\rangle)\}$, which is what is referred to as the $\alpha-$SRE:

$$M_\alpha(|\psi\rangle) = (1-\alpha)^{-1} \log \sum_{P \in \mathcal{P}_n} \Xi_P^\alpha(|\psi\rangle) - \log d \tag{30}$$

387 The SREs can be proved to be good magic monotones. Indeed, they are faithful to pure stabi-
388 lizer states, non-increasing under Clifford operations and additive with respect to the tensor
389 product.
390 In Ref. [111], an additional magic monotone is defined, the stabilizer linear entropy:

$$M_{lin}(|\psi\rangle) = 1 - d \sum_{P \in \mathcal{P}_n} \Xi_P^2(|\psi\rangle) \tag{31}$$

391 This measure is also faithful to pure stabilizer states and stable under Clifford operations.
392 However, it is not additive with respect to tensor product structure. It can be shown that the
393 following upper bound is true for the stabilizer linear entropy: $M_{lin}(|\psi\rangle) < 1 - 2(d+1)^{-1}$.
394 Throughout the manuscript we will be computing the 2nd-SRE $M_2$ and its density in the num-
395 ber $L$ of spins $m_2 = M_2/L$ for each considered spin chain.

396 The fact that SREs do not require minimization immediately opens up a plethora of pos-
397 sibilities in terms of their practical computation. Indeed, over the last 3 years, it has been
398 shown that SREs can be computed efficiently in tensor networks [59–63] as well as in gaus-
399 sian states [117], or via sampling in Monte Carlo methods [62, 71, 118]. We will mention and
400 utilize some of these techniques below.

### 3.3.2  SREs in spin models

The first model to be studied with respect to the 2-SRE was the *Transverse Field Ising Model* (TFIM), in Ref. [119]. In this work, the authors compute the 2-SRE in terms of the ground state correlation functions of the model. In the same paper, it is shown how to extract the SRE from just a few (even a single) spin measurements when the system is in its gapped phase. In this phase, in fact, the ground state of the TFIM possesses a well-localized 2-SRE, which is extractable via measurements on small subsystems. In the same work, it was shown that the 2-SRE of the ground state exhibits linear scaling in the system size and always peaks at the critical point of the model. The latter theoretical results were numerically confirmed in Ref. [60] using a *perfect Pauli-sampling* method, in Ref. [58] using the MPS replica-trick method, in Ref. [62] using a statistical approach based on a Markov chain sampling of Pauli strings and in Ref. [118] using a non-equilibrium Quantum Monte Carlo (QMC) approach. The model was also studied utilizing full replicated MPS in Ref. [58]. Moreover, in Ref. [120] the authors investigated the Pauli spectrum and the *filtered* SRE in the context of disordered TFIM, showing that both quantities can distinguish ergodic and many-body localization regimes.

The spin-1 XXZ model has been studied in Ref. [62, 70] both in terms of the full-state SRE and the so-called *long range magic*, which quantifies the magic originated by correlations between spatially separated subsystems. It is shown that the full-state SRE is rather featureless at the critical point, while the long-range magic actually identifies the phase transitions.

In Ref. [62], the authors also consider a $\mathbb{Z}_2$ lattice gauge theory which is dual to a 2D TFIM. Here they find that magic not only identifies the confinement-deconfinement transition, but also displays critical scaling behavior. The 2D TFIM was also considered in Ref. [118], where they adopt the previousy mentioned QMC approach and reproduce the MPS-based numerical results. Lattice gauge theories have been also considered in Ref. [121], where authors investigate the 2-SRE across the full phase diagram of the lattice Schwinger model in its Fendley-Sengupta-Sachdev spin-$\frac{1}{2}$ formulation. Magic is shown to be extensive throughout all the phases of the model, although ordered phases exhibit lower magic, while disordered phases exhibit higher magic. Again, the magnitude of the 2-SRE does not generally pinpoint phase transitions, but critical points can be effectively identified by peaks in the derivatives.

Another class of many-body Hamiltonians, known as *Rokhsar-Kivelson* models, allows for a *Stochastic Matrix Form* (SMF) decomposition. These systems' ground state has an analytic description throughout the entire phase diagram. Because of this, in Ref. [122] authors studied SREs in this class of systems, showing that they are relatively smooth across quantum phase transitions, although they become singular in their derivatives depending on the order of the transition. The SRE is also shown to reach its maximum at a cusp away from the critical point, where its derivative abruptly changes sign.

In [123], the entanglement complexity of *Rokhsar-Kivelson*-sign wavefunctions was investigated through the study of various properties, including SRE, entanglement entropy, and Entanglement Spectrum Statistics (ESS) [85, 124, 125]. The analysis revealed a transition in entanglement complexity driven by the sole parameter $\lambda$ of this family of states. This transition is detected by the 2-SRE and the critical point separates a phase characterized by volume-law entanglement scaling and universal (Wigner-Dyson) ESS from a phase exhibiting area-law entanglement scaling and non-universal ESS, a behavior that closely resembles that of high-energy excited states of the disordered XXZ model [126].

Due to their significance in QMB theory, topologically frustrated spin systems have also been analyzed in terms of nonstabilizerness. The work that fostered this line of research is Ref. [69]. Here, the authors study spin-$\frac{1}{2}$ chains with topological frustration and show that near a classical point, the ground state of these models can be mapped to a W-state via Clifford operations. They also find that, in these systems, the SRE results from a dominant local contribution and a subdominant one linked to the delocalized nature of W-states. In particular,

non-frustrated systems near their classical point approach a GHZ-state with zero SRE. Frustrated 1D systems, instead, possess a 2-SRE that is given by an extensive term (also found in non-frustrated 1D systems) and a logarithmic term that cannot be resolved by local quantities. Topological frustration has also been considered in Ref. [127], where the SRE has been proven to be sensitive to transitions from zero to finite momentum ground states in the frustrated XYZ model. Their results reveal a discontinuity in the SRE induced by topological frustration, marking the first occurrence where such a transition can be detected solely through the SRE.

Moreover, the nonstabilizernes was also studied in the context of quench dynamics. In fact, understanding how magic resources build up and propagate in many-body quantum systems emerges as a fundamental question, with a potential impact on current and near-term quantum devices. More recently, random quantum circuits allowed to resolve the evolution of magic resources. Within random quantum circuits, the nonstabilizerness exhibits similar phenomenology, reaching the stationary value of typical many-body states in a timescale that is logarithmic in the system size [128]. Significantly, magic spreading differs from the ballistic increase of the entanglement entropy, whose saturation timescale is linear in the system size. Since their introduction, the success of random quantum circuits is tied with arguments of typicality, leading to the widespread belief that they capture the qualitative features of nearly all ergodic quantum systems. In this context, several works challenge that assumption. In Ref. [129], the authors studied the behavior of magic in a QMB system after a sudden quench in an integrable model (more specifically, they consider the TFIM). They proved that the 2-SRE grows and eventually equilibrates over a time that scales at most linearly in the system size. Additionally, a stabilizer entropy length was defined to characterize the localization of the stabilizer entropy in subsystems. This length was shown to grow linearly with time, indicating that the 2-SRE spreads ballistically through the system until it becomes fully delocalized. The 2-SRE has also been studied in the Floquet dynamics of a *Kicked Ising Model*(KIM) and in the Hamiltonian dynamics of the *mixed fields Ising Model* (MIFM) in [128], where authors prove a difference in the scaling of the SREs in the two models. In particular, Floquet systems exhibit a scaling of the SRE anticoncentration and saturation in time that is logarithmic in the size of the system, while for Hamiltonian systems the scaling has been proven to be linear. In [130], the authors analyzed the dynamics of SRE and antiflatness in both integrable and non-integrable systems. They demonstrated that in free-fermion models, SRE and antiflatness exhibit a significant gap from their respective Haar values, reflecting the absence of complex quantum behavior. Conversely, in non-integrable and, more generally, non-free-fermion models, these quantities approach the Haar values closely. Therefore, in the long-time limit, SRE and antiflatness serve as indicators of whether a system can be effectively mapped to a free-fermion model.

# 4 Nonstabilizerness and entanglement structures in spin-1/2 models

As explained above, understanding quantum complexity in spin-1/2 systems requires exploring the interplay between nonstabilizerness and entanglement properties across different quantum phases. Spin-1/2 models provide a rich platform to investigate these phenomena due to their well-understood phase diagrams and diverse ground-state properties. By analyzing these models, we gain insights into how quantum complexity arises in various phases of matter and its connections to entanglement spectral quantities[3].

In this section, we focus on the relationship between magic and entanglement spectrum

---

[3]During the remainder of the manuscript, all the considered entanglement spectral quantities, if not stated otherwise, have been computed at half chain.

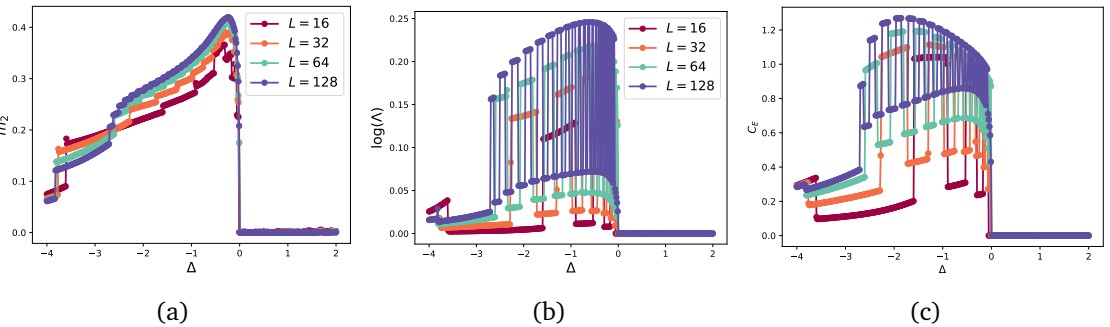

Figure 3: **SRE and antiflatness in the XXZ model**: (3a) SRE vs the interaction $\Delta$ for $h_z = 0.5$. (3b-3c) The antiflatness $\log(\Lambda)$ and $C_E$ as a function of the interaction $\Delta$ for $h_z = 0.5$.

antiflatness across several paradigmatic spin-1/2 models. Each subsection examines a specific model or class of models, highlighting how their ground-state properties reflect signatures of quantum complexity. We emphasize that all numerical simulations in this work were performed using tensor network methods. Since the estimation of the SRE is a less conventional task in the quantum many-body community, we provide a detailed account of the adopted numerical techniques in Appendix A.

We begin with the XXZ model, a cornerstone of condensed matter theory, known for its integrable structure and wide applicability. The model reduces to the XY chain for $\Delta = 0$, which is pivotal for understanding equilibrium phase transitions and serves as a foundational model for quantum information applications. We examine the role of magic and antiflatness as the system transitions from antiferromagnetic to paramagnetic phases under a transverse field, including the effects of factorized ground states at specific parameter values. The introduction of Dzyaloshinskii-Moriya interactions further enriches the phase diagram of the XY model, introducing chiral phases and modifying entanglement properties.

Finally we present the results for the Cluster Ising and Cluster XY Models. These models are of particular interest due to their ability to host topological phases and unconventional order. We explore the magic and anti-flatness in their ground states, emphasizing their distinct phase diagrams and potential implications for quantum technologies. The interplay between entanglement and nonstabilizerness in these systems sheds light on their computational and informational capabilities.

## 4.1 Heisenberg model

We consider the anisotropic Heisenberg chain described by the following Hamiltonian:

$$H = -\frac{J}{2}\sum_{j=1}^{L}\left\{\frac{1-\gamma}{2}\sigma_j^x\sigma_{j+1}^x + \frac{1+\gamma}{2}\sigma_j^y\sigma_{j+1}^y + \frac{\Delta}{2}\sigma_j^z\sigma_{j+1}^z\right\} - h\sum_{j=1}^{L}\sigma_j^z \tag{32}$$

where $L$ is the number of spins in the chain. The model has antiferromagnetic (AFM) exchange coupling $J > 0$, anisotropy $\Delta$, and uniform magnetic field $h$ acting on $\sigma_j^z$.

Considering the Hamiltonian in Eq. 32 with $\gamma = 0$, one obtains the XXZ spin chain with nearest neighbor interaction, which is integrable and can be analytically solved via Bethe ansatz [131]. The XXZ parameter space hosts three phases: ferromagnetic (FM), spin-liquid (XY), and antiferromagnetic (AFM). These three phases are separated by two lines $h_s = \frac{J}{2}(1-\Delta)$ and $h_c$. For $h = 0$ the XXZ model undergoes a first-order phase transition at $\Delta = 1$ and a Kosterlitz-Thouless infinite-order phase transition at $\Delta = -1$ [132]. The line between these two points is a critical line, where the excitation gap vanishes.

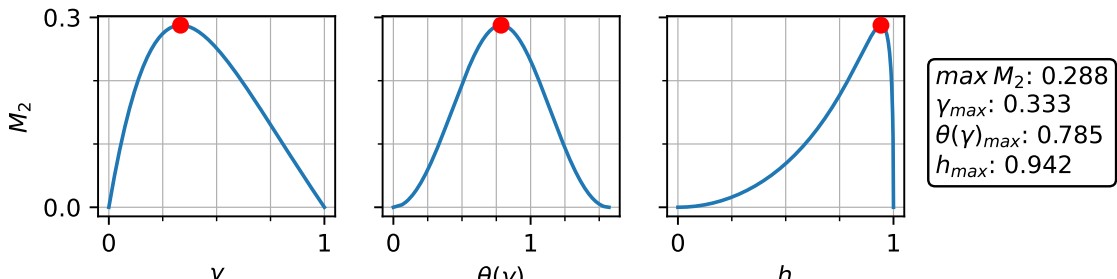

Figure 4: **SRE on the XY model's separability circle**: The 2-SRE ($M_2$) of the state $\left|\phi_i^{XY}\right\rangle$ (Eq. 38) with respect to $\gamma \in [0,1]$, $\theta(\gamma) = \arccos\sqrt{(1-\gamma)/(1+\gamma)}$ and $h = \sqrt{1-\gamma^2}$.

To analyze the quantum complexity of the Heisenberg chain, we examine the SRE, the antiflatness of the entanglement spectrum, and the capacity of entanglement ($C_E$) (with all spectral quantities computed in the middle of the spin chain).

Figure 3 illustrates these quantities as a function of $\Delta$ and at a fixed transverse field $h_z = 0.5$. In panel (a), the SRE shows distinct behavior across the different phases. In the ferromagnetic phase ($\Delta > 0$), the SRE is minimal, indicating that the ground state is close to a stabilizer state. As $\Delta$ decreases and the system enters the critical spin-liquid phase, the SRE grows, reaching a peak near the phase transition. This suggests that the nonstabilizerness of the ground state is maximized at criticality, consistent with the intuition that quantum complexity tends to be higher in critical systems (even if exceptions to this fact are known, see e.g. Ref. [62]). Panels (b) and (c) show that the antiflatness $\log(\Lambda)$ and the capacity of entanglement ($C_E$) share a similar trend: both measures peak at the transition between the spin-liquid and AFM phases, reinforcing their role as indicators of quantum criticality.

The scaling of these quantities with system size $L$ suggests that the observed peaks sharpen as $L$ increases, consistent with the expectation that quantum complexity measures exhibit critical singularities in the thermodynamic limit. We note that, differently from the SRE, both the antiflatness and capacity of entanglement feature strong parameter oscillations within the critical phase. This is likely reminiscent of the fact that the entanglement spectrum can change rather abruptly within the critical regime.

## 4.2 XY model

In this section, we consider the transverse field XY model, whose Hamiltonian reads:

$$H^{XY} = \frac{J}{2}\sum_{i=1}^{L-1}\left\{\frac{(1+\gamma)}{2}\sigma_i^x\sigma_{i+1}^x + \frac{(1-\gamma)}{2}\sigma_i^y\sigma_{i+1}^y\right\} - h\sum_{i=1}^{L}\sigma_i^z \tag{33}$$

where the spin-spin interaction terms do not involve the first and the last spins of the chain, so we are in Open Boundary Conditions (OBC). Here, we consider $\gamma \geq 0$, $J = 1$ and $L$ being even. Our aim is to characterize the ground state of the model in terms of quantum complexity. Specifically, we will study its entanglement, nonstabilizerness, and the two quantum complexity measures introduced in Sec. 2: the *antiflatness of the entanglement spectrum* ($\mathcal{F}$) and the *capacity of entanglement* ($C_E$). In particular, we adopt the von Neumann entropy ($\mathcal{S}$) as the measure for the entanglement, while as a measure of nonstabilizerness we choose the 2-SRE ($M_2$). We consider the system as bipartite, with each partition containing exactly half of the spins in the chain, arranged contiguously. Unless stated otherwise, our analysis of $\mathcal{S}$, $\mathcal{F}$, and $C_E$ will focus on one half of the chain. We begin by examining these quantities on the separability circle. We, then, explore their full phase diagrams with respect to $\gamma$ and $h$.

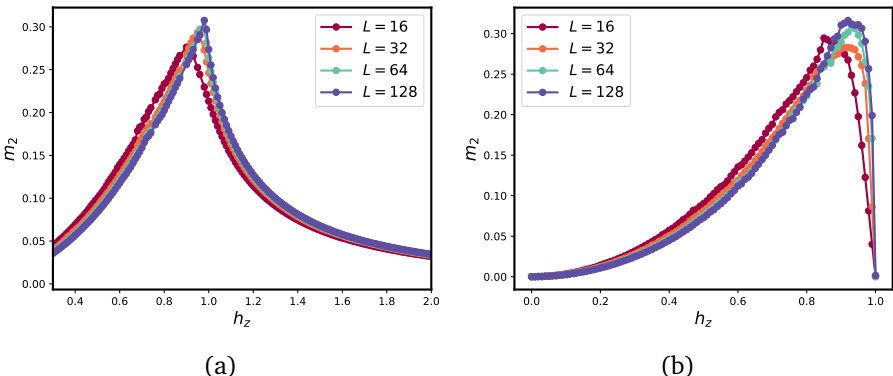

Figure 5: **SRE in the XY model**: (5a) SRE vs the magnetic field $h_z$ for $\gamma = 0.7$. (5b) SRE vs the magnetic field $h_z$ along the separable line $\gamma = \sqrt{1-h_z^2}$.

### 4.2.1 Separability circle

When the anisotropy parameter $\gamma$ takes a value in the interval $[0,1]$, the quantum XY model admits a transverse field intensity $h_{sep} = \sqrt{1-\gamma^2}$ at which the GS is fully-factorized[4]. The line of equation $h^2 + \gamma^2 = 1$, where the GS becomes fully factorized, is often referred to as *separability circle* [133]. Here, the GS assumes the following expression:

$$\left|GS^{XY}\right\rangle = \bigotimes_i \left|\phi_i^{XY}\right\rangle \tag{34}$$

These states, also known as *Classical-like Ground States (CGS)*, are fully separable quantum states, meaning they exhibit no quantum entanglement. This means that CGS possess vanishing entanglement entropies, and, by the same token, a vanishing antiflatness $\mathcal{F}$ and capacity of entanglment $C_E$. The separability circle thus plays a key role in identifying the limits of quantum behavior in the XY model, motivating a further characterization of CGS in terms of nonstabilizerness.

The CGS in the quantum XY model are obtained by the tensor product of single-qubit states [133]:

$$\left|\phi_i^{XY}\right\rangle = (-1)^i \cos\frac{\theta_\gamma}{2}\left|\downarrow_i\right\rangle + \sin\frac{\theta_\gamma}{2}\left|\uparrow_i\right\rangle, \quad \cos\theta_\gamma = \sqrt{\frac{1-\gamma}{1+\gamma}} \tag{35}$$

Considering the expression of the SRE for pure states (Eq. 30), and exploiting its additivity with respect to the tensor product, we can write the $\alpha$-SRE of the full GS as:

$$M_\alpha\left(\left|GS^{XY}\right\rangle\right) = L M_\alpha\left(\left|\phi_i^{XY}\right\rangle\right) \tag{36}$$

where the $\alpha$-SRE of the single qubit states in Eq. 35 has the following expression:

$$M_\alpha\left(\left|\phi_i^{XY}\right\rangle\right) = (1-\alpha)^{-1}\log\left[\frac{1}{d^\alpha}\left(1 + \sin^{2\alpha}(\theta_\gamma) + \cos^{2\alpha}(\theta_\gamma)\right)\right] - \log d \tag{37}$$

We are interested in the 2$-$SRE, which assumes the following expression when computed for the CGS in Eq. 34:

$$M_2\left(\left|\phi_i^{XY}\right\rangle\right) = -\log\left[\frac{1}{4}\left(1 + \sin^4(\theta_\gamma) + \cos^4(\theta_\gamma)\right)\right] - \log 2 \tag{38}$$

---

[4]At finite size, the OBC ground state is not exactly separable on the separability circle, as weak boundary entanglement persists. Nonetheless, OBC and PBC ground states coincide in the thermodynamic limit, with differences in extensive quantities scaling as $\mathcal{O}(1/N)$. Also, the 2-SRE of the OBC ground state already converges to that of the PBC case at moderate system sizes (cf. Fig. 5).

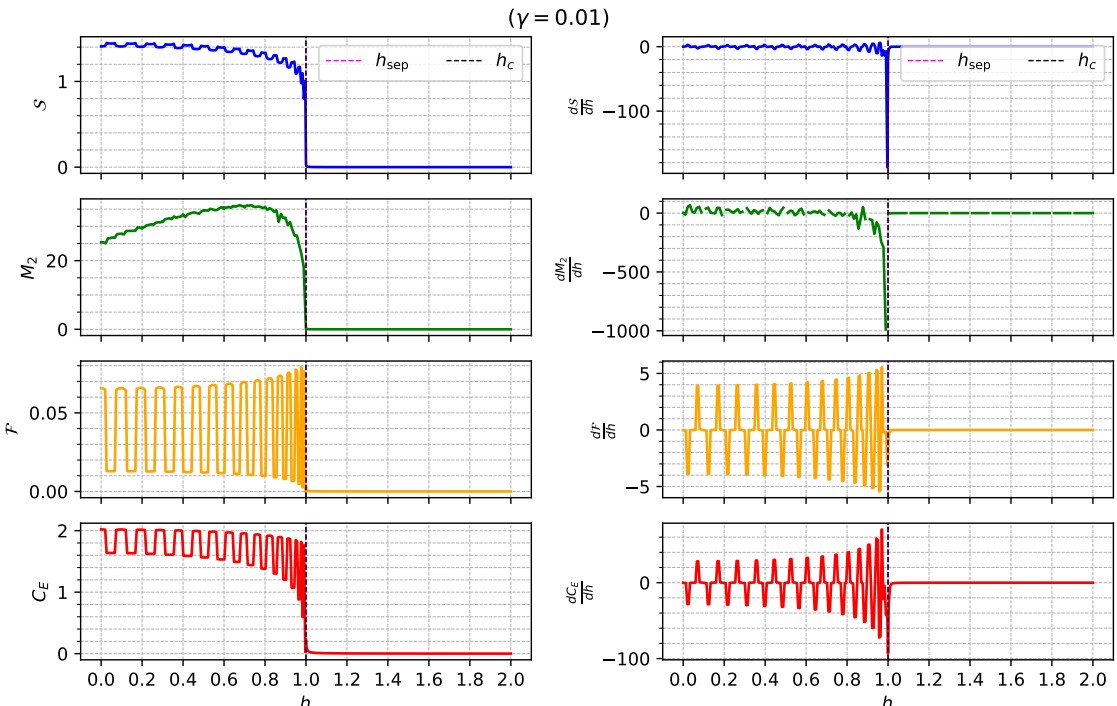

Figure 6: **Entanglement, nonstabilizerness, antiflatness and capacity of entanglement of the TFXY's GS at** $\gamma = 0.01$, $L = 64$: Comparative plots of the von Neumann entropy ($\mathcal{S}$), antiflatness ($\mathcal{F}$), 2-SRE ($M_2$), and capacity of entanglement ($C_E$) of the ground state (GS) of the quantum XY model, along with their derivatives with respect to the transverse field intensity $h$, for a fixed anisotropy parameter $\gamma = 0.01$. The $\mathcal{S}, \mathcal{F}$, and $C_E$ are computed at half of the chain.

We plot the behavior of $M_2$ with respect to $\theta_\gamma$ and $\gamma$ in Fig. 4, where we see that $M_2$ peaks at $\theta_\gamma = \frac{\pi}{4}$, at which the GS achieves the magic of a $T$−state. In Fig. 5b, $M_2$ is computed along the separable line: here we observe complete agreement with Fig. 4.

### 4.2.2 Quantum Complexity in the full parameter space

In Figure 6-9 we presents some plots, with the intent of providing a visual tool for inspecting what happens to entanglement ($\mathcal{S}$), nonstabilizerness ($M_2$), the antiflatness of the entanglement spectrum ($\mathcal{F}$) and the capacity of entanglement ($C_E$) in the GS of the XY model across the entire parameter space at a finite size of the system which is fixed at $L = 64$ spins.

In Figure 6-7 we take horizontal slices of the phase diagram by fixing, for each plot, a value of the anisotropy parameter $\gamma$ and considering a transverse field intensity $h \in [0, 2]$. Here, numerical simulations reveal a complex interplay between $S$ and $M_2$. Inside the spearability circle, for low values of $\gamma$, $M_2$ exhibits oscillations that closely resemble those of $\mathcal{S}$, as evidenced by the similar behavior of their respective derivatives, $\frac{dM_2}{dh}$ and $\frac{d\mathcal{S}}{dh}$. This correspondence is especially pronounced for shorter chain lengths, where the oscillations in $\mathcal{S}$ are more prominent and the period of the oscillations is greater. In this low-$\gamma$ regime, the antiflatness $\mathcal{F}$ and the capacity of entanglement $C_E$ display strikingly similar oscillatory patterns, which we will discuss in a moment. At this point, it is important to recall that the antiflatness $\mathcal{F}$ is a spectral quantity of the reduced state, and it is zero for both product states and stabilizer states [108]. Because of it being a spectral quantity, the antiflatness is also invariant under bipartite unitary transformations. This implies that local changes of basis cannot either increase

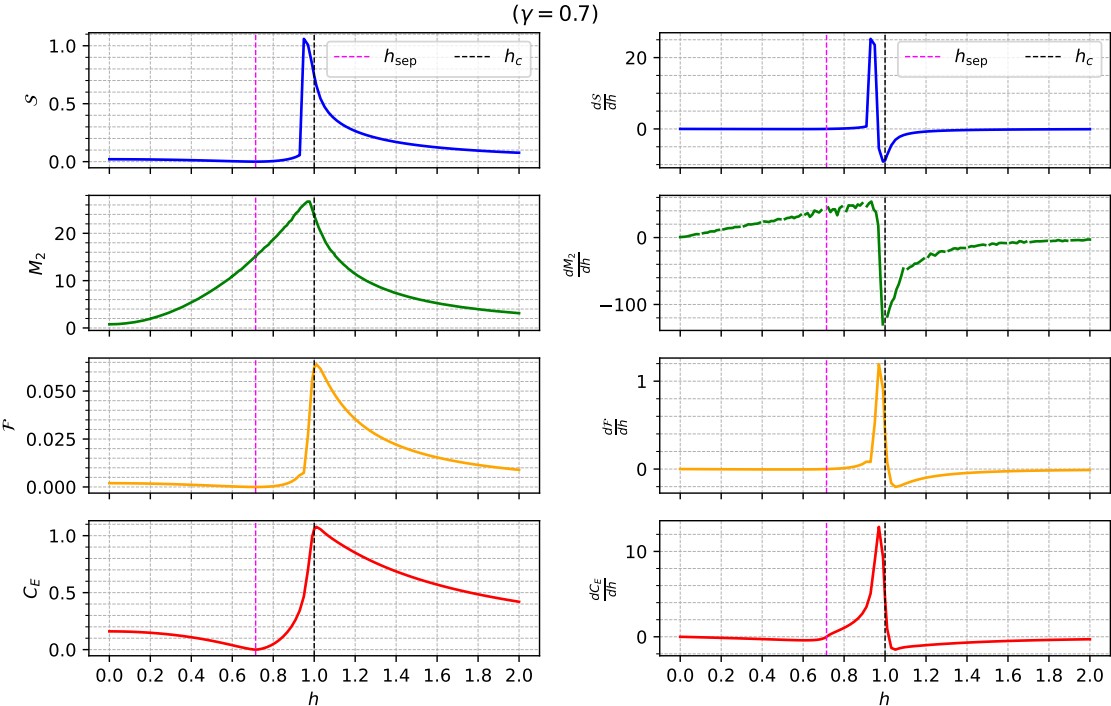

Figure 7: **Entanglement, nonstabilizerness, antiflatness and capacity of entanglement of the TFXY's GS at** $\gamma = 0.7$, $L = 64$ :Comparative plots of the von Neumann entropy ($\mathcal{S}$), antiflatness ($\mathcal{F}$), 2-SRE ($M_2$), and capacity of entanglement ($C_E$) of the ground state (GS) of the quantum XY model, along with their derivatives with respect to the transverse field intensity $h$, for a fixed anisotropy parameter $\gamma = 0.7$. The $\mathcal{S}, \mathcal{F}$ and $C_E$ are computed at half of the chain.

or decrease the antiflatness of a state. Consequently, a state with non-vanishing entanglement and magic can only exhibit vanishing antiflatness if the magic originates solely from local basis changes within the subsystems, implying that the magic is localized. This behavior is what we observe in the low-$\gamma$ regime, where oscillations in $\mathcal{F}$ occur in a region of the parameter space where both $\mathcal{S}$ and $M_2$ are non-zero. These oscillations reflect, then, a periodic transition between localized and non-localized magic within the considered bipartition of the system. Increasing $\gamma$, as in Fig. 7, we shift towards a region of the phase diagram where magic is well localized in the ordered phase of the model, due to the low value of entanglement, and starts to spread in the bipartition after the critical point.

Let us make an additional remark about Figure 6-7. From these, we see that, even at a finite size, as already pointed out in literature, although the $M_2$ of the full ground state peaks away from the critical point $h_c = 1$, its first derivative becomes singular at the latter. Numerical simulations in Fig. 5 also reveal that the peak of $M_2$ is shifted away from the critical point only for finite sizes of the system, while it is expected to coincide with the critical point in the thermodynamic limit. Notably, instead, the antiflatness $\mathcal{F}$ and the capacity of entanglement $C_E$, consistently peak at the critical point even at finite sizes, proving to be effective at capturing the phase transition.

Figure 9 presents the phase diagrams of $\mathcal{S}$, $M_2$, $\mathcal{F}$, and $C_E$ as functions of the transverse field $h$ and the anisotropy parameter $\gamma$, allowing us to explore their behavior across a broader range of $\gamma$ values. In the low-anisotropy regime, these quantities exhibit distinct oscillatory patterns, corresponding to the magic localization phenomenon, with $\mathcal{S}$, $\mathcal{F}$, and $C_E$ showing pronounced fluctuations, while $M_2$ undergoes noticeably smoother ones. Notably, $M_2$ appears

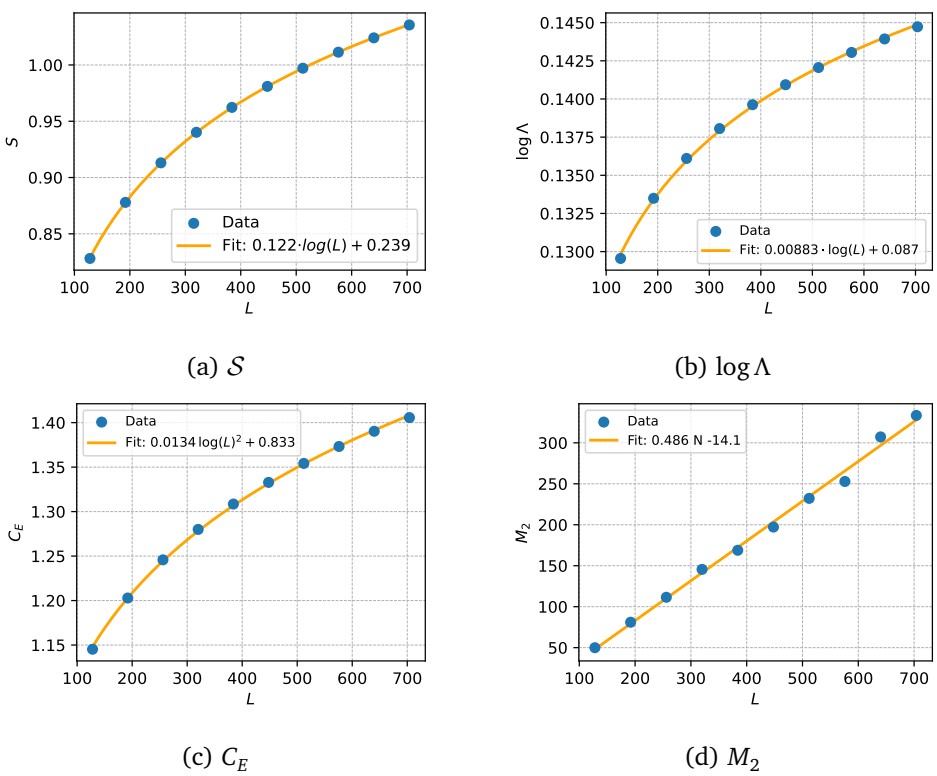

Figure 8: **Finite size scaling in the XY model**: Scaling at criticality in the GS of the TFXY model at $\gamma = 0.7$ for (8a) the Von Neumann entropy $S$, (8b) the $\log \Lambda$, (8c) capacity of entanglement $C_E$ and (8d) 2-SRE $M_2$. Here, $S$, $\mathcal{F}$, $\log \Lambda$ and $C_E$ are computed at half of the chain (i.e. for a block of $L/2$ contiguous spins), while $M_2$ is computed for the full GS.

largely unaffected by the separability circle. For higher values of $\gamma$, at the peak of entanglement, both the antiflatness $\mathcal{F}$ and the capacity of entanglement $C_E$ reach a local maxima. This behavior is a finite-size effect, as the peaks are expected to vanish in the thermodynamic limit.

The magic-localization phenomenon observed in Fig. 6 and Fig. 9 originates from the well-known parity oscillations exhibited by the finite-size ground state of the XY model. As discussed in detail in Ref. [134], within the separability circle the ground state of a finite chain is non-degenerate and alternately coincides with one of the two parity-broken thermodynamic ground states. As the transverse field $h$ is varied, the parity of the finite-size ground state switches periodically, producing corresponding oscillations in all quantum resources, including nonstabilizerness and entanglement-spectral indicators. As the anisotropy parameter $\gamma$ increases, both the energy splitting between the two parity-broken sectors and the associated oscillations in their quantum resources become progressively suppressed. This explains why the oscillatory behavior in Fig. 9 is most pronounced close to the isotropic limit of the model and gradually fades away deeper in the anisotropic regime.

In addition to the phase diagrams, in Fig. 8 we provide the finite-size scaling of the previously discussed quantities. The Von Neumann entropy (and, more in general, the $\alpha$-Rényi entropies) is proven to scale logarithmically in the size of the considered subsystem for the GS of the TFXY model at criticality [7, 135]:

$$S_\alpha(\rho_A) = \frac{1}{1-\alpha} \log Tr(\rho_A^\alpha) = \mathcal{O}(\log L) \tag{39}$$

The scaling of the antiflatness can be inferred from the behavior of the $\alpha$-Rényi entropies.

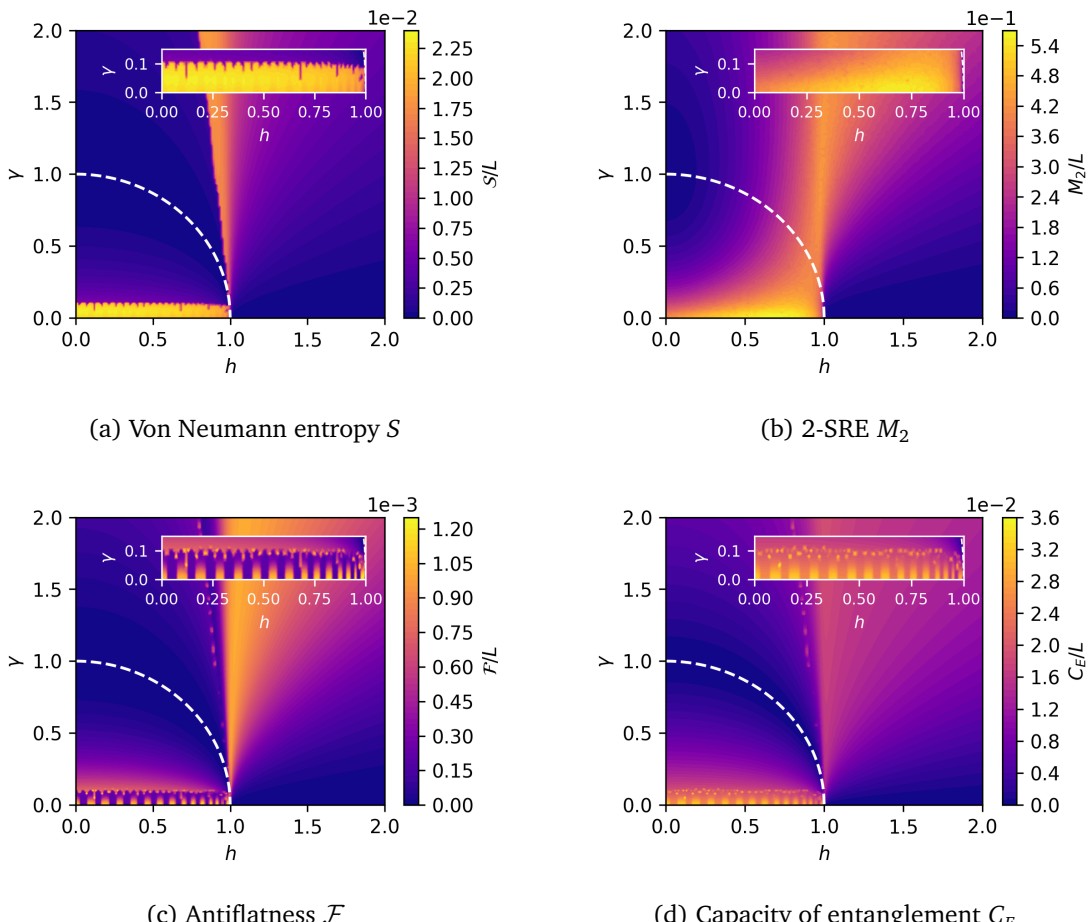

(a) Von Neumann entropy $S$

(b) 2-SRE $M_2$

(c) Antiflatness $\mathcal{F}$

(d) Capacity of entanglement $C_E$

Figure 9: **Entanglement, nonstabilizerness, antiflatness and capacity of entanglement phase diagrams of the TFXY model (L=64):** Phase diagrams of (9a) von Neumann entropy $S$, (9b) 2-SRE $M_2$, (9c) antiflatness of the entanglement spectrum $\mathcal{F}$, and (9d) capacity of entanglement $C_E$ as functions of $h$ and $\gamma$ in the quantum XY model with $L = 64$ spins and OBC. Notice that all the quantities have been normalized in the number of spins $L$, and that $\mathcal{S}, \mathcal{F}$ and $C_E$ have been computed at half of the chain.

Specifically, we have

$$\mathcal{F}(\rho_A) = e^{-2S_3(\rho_A)} - e^{-2S_2(\rho_A)}, \tag{40}$$

and since both $S_2$ and $S_3$ scale logarithmically with the subsystem size, it follows that the antiflatness exhibits a power-law scaling with $L$.

In Fig. 8b we plot the scaling of the log-ratio $\log \Lambda$ (defined in Eq. 14). This quantity exhibit a logarithmic scaling in the system size, which is expected since we can write:

$$\log \Lambda_A = 2(S_2(\rho_A) - S_3(\rho_A)) \tag{41}$$

and since Rényi entropies scale as $\mathcal{O}(\log L)$, the log-ratio $\log \Lambda$ will also scale as $\mathcal{O}(\log L)$. In Fig. 8c we also plot the finite size scaling of the capacity of entanglement $C_E$. Here, we see it scaling as $\mathcal{O}(\log^2 L)$. We can check this scaling with some manipulations, starting with rewriting $C_E$ as:

$$C_E(\rho_A) = Tr(\rho_A \log^2 \rho_A) - Tr^2(\rho_A \log \rho_A) = Tr(\rho_A \log^2 \rho_A) - S^2(\rho_A) \tag{42}$$

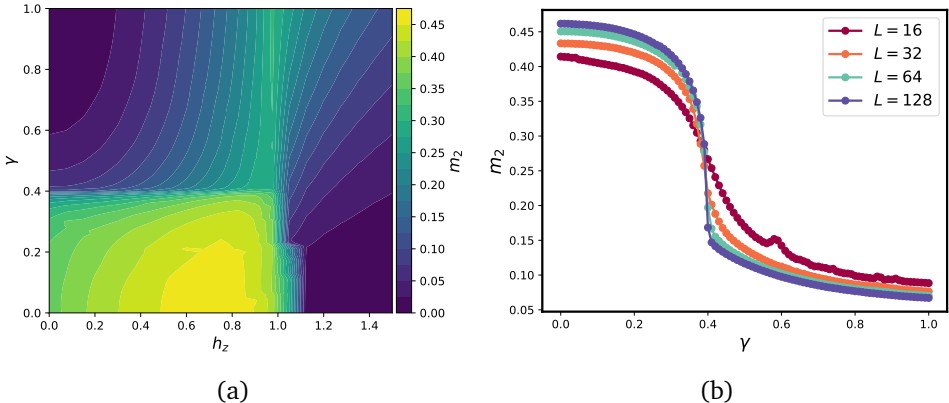

Figure 10: **Phase diagram of SRE for the XY model with Dzyaloshinskii-Moriya interaction:**(10a) Contour plot of SRE as a function of $\gamma$ and $h_z$ for $D = 0.2$. (10b) Density of SRE as a function of $\gamma$ for $h_z = 0.4$ and $D = 0.2$.

In the previous expression we already know the scaling of the second term, which is $\mathcal{O}(\log^2 L)$. We, then, focus on the first term. Let us notice that:

$$\rho_A^\alpha = e^{\log \rho_A^\alpha} = e^{\alpha \log \rho_A} \tag{43}$$

The derivatives of $\rho_A^\alpha$ take, therefore, the following expression:

$$\frac{\mathrm{d}^n \rho_A^\alpha}{\mathrm{d}\alpha^n} = \rho_A^\alpha \log^n \rho_A \tag{44}$$

from which:

$$Tr(\rho_A \log^2 \rho_A) = Tr\Big(\frac{\mathrm{d}^2 \rho_A^\alpha}{\mathrm{d}\alpha^2}\Big|_{\alpha=1}\Big) \tag{45}$$

From Eq. 39 it follows that:

$$Tr(\rho_A^\alpha) = \mathcal{O}(L^{1-\alpha}) \tag{46}$$

and using the linearity of the trace we write:

$$\frac{\mathrm{d}^2 Tr(\rho_A^\alpha)}{\mathrm{d}\alpha^2}\Big|_{\alpha=1} = \frac{\mathrm{d}^2 \mathcal{O}(L^{1-\alpha})}{\mathrm{d}\alpha^2}\Big|_{\alpha=1} = \mathcal{O}\big(\log^2(L)\, L^{1-\alpha}\big)\Big|_{\alpha=1} = \mathcal{O}(\log^2 L) \tag{47}$$

which confirms the scaling mentioned above. Finally, in Fig. 8d we see the 2-SRE scaling linearly in the system size ($\mathcal{O}(L)$), which is compliant with the results obtained in [119].

## 4.3 XY model with Dzyaloshinskii-Moriya interaction

We consider the spin chain described by the following Hamiltonian

$$H = J \sum_{j=1}^N \left\{ \frac{1-\gamma}{2} \sigma_j^x \sigma_{j+1}^x + \frac{1+\gamma}{2} \sigma_j^y \sigma_{j+1}^y + \overline{D}\big(\overline{\sigma}_j \times \overline{\sigma}_{j+1}\big) \right\} - h \sum_{j=1}^N \sigma_j^z \tag{48}$$

where $N$ is the number of spins in the chain. The model has antiferromagnetic (AFM) exchange coupling $J > 0$, Dzyaloshinskii-Moriya interaction with strength $\overline{D}$, and uniform magnetic field $h$ acting on $\sigma_j^z$. Here we presume that the $\overline{D}$ vector is along the direction perpendicular to the plane, i.e., $\overline{D} = D_z$. The parameter $\gamma$ measures the anisotropy of spin-spin interactions in the xy plane which typically varies from 0 (isotropic XY model) to 1 (Ising model).

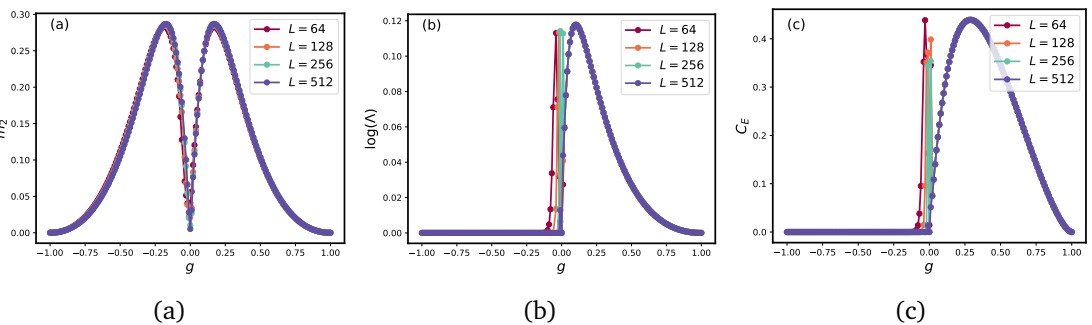

Figure 11: **SRE and antiflatness in the Cluster Ising model:**(11a) SRE vs $g$. As shown in the plot the SRE peaks at the point $g = \pm(3-2\sqrt{(2)})$. (11b-11c antiflatness $\log(\Lambda)$ and $C_E$ as a function of $g$. Both quantities vanish in the topological phase and show a peak only at the criticality.

To analyze the role of nonstabilizerness in the XY model with Dzyaloshinskii-Moriya (DM) interaction, $D \neq 0$, we compute the SRE across different interaction regimes. The contour plot in the left panel of Fig. 10 shows the SRE as a function of the anisotropy parameter $\gamma$ and the transverse field $h_z$ with a fixed DM interaction strength $D = 0.2$. The right panel presents the finite-size scaling of the SRE as a function of $\gamma$, for different system sizes $L$. From the contour plot, we observe that the SRE is generally maximized in the critical region where $\gamma$ is small and $h_z$ is moderate, roughly around $\gamma = 0.4$ and $h_z = 0.8$. This suggests that in this region, the ground state exhibits the highest degree of nonstabilizerness, potentially due to strong quantum fluctuations induced by competing interactions. Increasing $h_z$ the system eventually transitions to a paramagnetic phase where the SRE decreases sharply. Additionally, for large anisotropy $\gamma \to 1$, the system approaches the Ising limit, where stabilizer properties become dominant and the SRE is significantly reduced. The right panel further confirms this trend by displaying the finite-size scaling of the SRE as a function of $\gamma$. We see that for small $\gamma$, the SRE saturates to a finite value even for large $L$, suggesting a robust presence of nonstabilizerness in this regime. However, as $\gamma$ increases, the SRE decreases monotonically, converging to a smaller but finite value at large $L$. This indicates that while the Ising limit still retains some nonstabilizerness, its magnitude is significantly lower than in the more isotropic XY-like regime.

These results highlight the interplay between external fields and the DM interaction in shaping the stabilizer properties of the system. The numerical analysis indicates a clear shift in the behavior of the SRE, underscoring the sensitivity of these measures to chiral phases induced by the DM interaction.

## 4.4 Cluster Ising model

A *Symmetry-Protected Topological* (SPT) phase is characterized by the presence of a protecting symmetry that prevents the adiabatic deformation of the system's ground state into a trivial product state. Unlike conventional phases, SPT phases cannot be described using local order parameters and are instead diagnosed using nonlocal observables, such as string order parameters.

We study a class of 1D spin-1/2 chain models with the following Hamiltonian:

$$\hat{H} = \sum_j \left\{ -g_{zz}\sigma_j^z\sigma_{j+1}^z - g_x\sigma_j^x + g_{zxz}\sigma_j^z\sigma_{j+1}^x\sigma_{j+2}^z \right\}. \tag{49}$$

This model exhibits both global spin-flip symmetry (generated $\prod_j \sigma_j^x$) and time-reversal symmetry, which together protect a $\mathbb{Z}_2 \times \mathbb{Z}_2$ SPT phase. The system also supports a trivial phase and a symmetry-broken phase.

We focus on a solvable trajectory in parameter space defined by the relations $g_{zz} = 2(1-g^2)$, $g_x = (1+g)^2$, and $g_{zxz} = (g-1)^2$. Along this path, the ground state admits an exact MPS description with bond dimension two [136]. The corresponding tensors are given by

$$A^{(1)} = \frac{1}{\sqrt{1+|g|}} \begin{bmatrix} 1 & sign(g)\sqrt{|g|} \\ 0 & 0 \end{bmatrix} \qquad A^{(0)} = \frac{1}{\sqrt{1+|g|}} \begin{bmatrix} 0 & 0 \\ \sqrt{|g|} & 1 \end{bmatrix} \tag{50}$$

This variational state captures the ground state physics across all three phases, including the tricritical point at $g = 0$, where three different phases meet. Notably, although the system is critical at $g = 0$, the entanglement entropy remains finite and bounded by $\log 2$, reflecting the absence of conformal symmetry.

The SRE associated with this MPS can be computed exactly using transfer matrix techniques for translation-invariant systems. As shown in [137], the resulting expression is:

$$m_2 = -\log\left(\frac{1 + 14g^2 + g^4}{(1+g)^4}\right) \tag{51}$$

This function displays a pronounced nonmonotonic behavior as $g$ varies, with local maxima at $g = \pm(3 - 2\sqrt{2})$, where the stabilizer entropy attains its peak value of approximately 0.3. These peaks reflect enhanced non-Clifford correlations in the ground state. In contrast, the SRE vanishes at $g = \pm 1$ and $g = 0$. For $g = 1$, the ground state is a trivial product state, while for $g = -1$, it corresponds to a cluster state, a well-known stabilizer state within the SPT phase. At $g = 0$, the system sits at the tricritical point and the thermodynamic ground state is a GHZ state [138], also a Clifford state with vanishing SRE.

Fig. 11 illustrates the dependence of $m_2$, the antiflatness measure $\log \Lambda$, and the entanglement capacity $C_E$ on the tuning parameter $g$. Data is shown for different system sizes to highlight size dependence. The first panel shows the behavior of the SRE. It presents a clear double-peak structure centered near $g = \pm(3 - 2\sqrt{2})$, with dips at the three stabilizer points discussed above. The second panel shows the antiflatness, which also peaks at $g = 0$ and $g = (3 - 2\sqrt{2})$, indicating that the departure from stabilizerness is correlated with an increased complexity in the ground state wavefunction. Finally, the third panel presents the entanglement capacity $C_E$, which displays a strong peak around the phase transition, further reinforcing that this region is where quantum correlations are maximized.

Notably, all three quantities sharpen with increasing system size, indicating the emergence of critical-like scaling near the transition. This underlines the utility of nonstabilizerness as a probe of quantum phase transitions, particularly in scenarios where standard symmetry-breaking order parameters are absent.

Moreover, the figure also reveals an important finite-size effect at the tricritical point $g = 0$: while, in the thermodynamic limit, the SRE vanishes due to the ground state being a GHZ state, for finite system sizes, we observe that $m_2$ remains finite. This indicates that the GHZ-like structure of the ground state is not perfectly realized at small $L$ but rather emerges asymptotically as $L \to \infty$. This behavior is expected, as the GHZ state exhibits a well-known finite-size crossover where entanglement and magic properties only become sharply defined in the infinite-size limit. This finite-size deviation is similarly reflected in the antiflatness and the entanglement capacity both of which also exhibit residual values at $g = 0$ for small $L$. These results underscore the necessity of considering large system sizes when analyzing magic and entanglement properties near phase transitions, as finite-size effects can obscure the asymptotic structure of the stabilizer content.

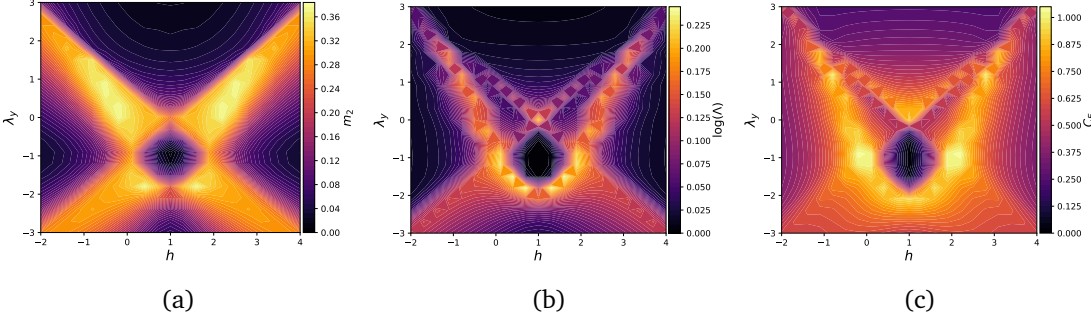

Figure 12: **Phase diagram of Cluster XY model:** We show the behaviour of stabilizer entropy $m_2$, antiflatness and entanglement capacity $C_E$ for the ground state of the Hamiltonian 52 for $\lambda_x = 0$ and for $L = 128$. (12a) Contour plot of $m_2$ as a function of $h$ and $\lambda_y$, (12b) Contour plot of $\Lambda$ as a function of $h$ and $\lambda_y$, (12c) Contour plot of $C_E$ as a function of $h$ and $\lambda_y$

## 4.5 Cluster XY model

The last model we consider is the cluster XY one, where cluster interactions complement the XY model in a transverse field. The Hamiltonian reads:

$$\hat{H} = \sum_{j=1}^{N} -\sigma_{j-1}^x \sigma_j^z \sigma_{j+1}^x - h\sigma_j^z + \lambda_y \sigma_j^y \sigma_{j+1}^y + \lambda_x \sigma_j^x \sigma_{j+1}^x \tag{52}$$

The model shows an interesting phase diagram with phases that appear because of the competition between the XY and cluster terms.

In the case of $\lambda_x = 0$, the Hamiltonian does not feature Ising interactions of type $\sigma_j^z \sigma_{j+1}^z$. However, two of the regions next to the cluster phase can be connected adiabatically, respectively, to ferromagnetic and antiferromagnetic states in the x direction. Something similar happens for $h = 0$. The Hamiltonian does not have a transverse term that tries to polarize all the spins in the same $z$ direction, nevertheless this phase is present in the reduced phase diagram.

By computing the SRE, the antiflatness of the entanglement spectrum, and the capacity of entanglement in the ground state of the Cluster-XY model, we are able to map out the critical regions and phase transitions in the model. Figure 12 illustrates the phase diagram as a function of the transverse field strength $h$ and the interaction parameter $\lambda_y$, providing insights into the different quantum phases and their respective transitions.

In the first panel of Figure 12, we plot the stabilizer Rényi entropy. The structure of the phase diagram reveals distinct regions of low and high nonstabilizerness, with a characteristic suppression of magic in the vicinity of $h = 1$. The central dark region corresponds to a phase with minimal nonstabilizerness, while the outer regions with bright intensity suggest highly non-Clifford states. This behavior aligns with the expectation that magic is generally amplified in quantum critical regions and suppressed in phases with local stabilizer structure.

In the second panel, we present the antiflatness of the entanglement spectrum, quantified via $\log(\Lambda)$. The observed pattern in $\log(\Lambda)$ closely resembles that of the SRE, reinforcing the notion that the entanglement spectral structure is deeply intertwined with nonstabilizerness. The bright lobes along the $\lambda_y$ axis indicate regions where the entanglement spectrum is highly structured, a feature that suggests a robustly ordered phase in the presence of dominant cluster interactions.

Finally, the third panel shows the capacity of entanglement, $C_E$, which quantifies the fluctuations of the entanglement entropy. Similar to the previous quantities, $C_E$ highlights critical

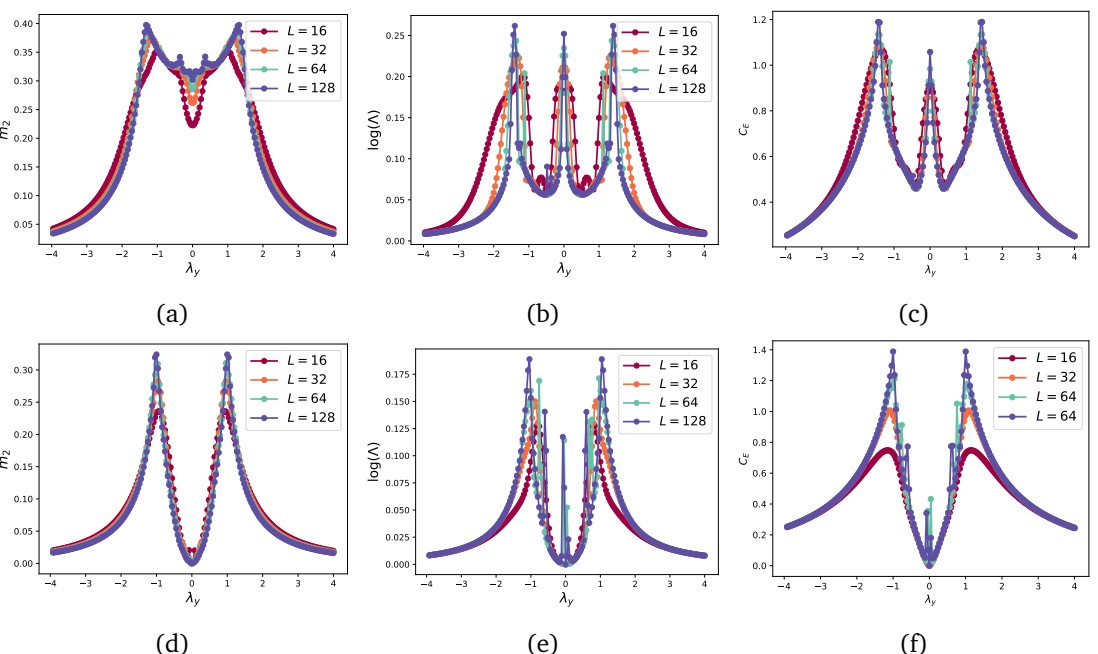

Figure 13: **Scaling of SRE, antiflatness $\Lambda$ in the Cluster XY model**: The top row (13a-13c) corresponds to $h = 0$, while the bottom row (13d-13f) corresponds to $h = 1$. In both cases, all three quantities exhibit peaks at the critical points, signaling phase transitions driven by the competition between the cluster interaction and the XY couplings. The peaks sharpen with increasing system size, consistent with the expected scaling behavior at quantum phase transitions. The structure of the peaks in the $h = 1$ case is simpler than in $h = 0$, indicating that the transverse field reduces the number of competing phases. These results highlight the role of stabilizer Rényi entropy, antiflatness, and entanglement capacity as effective probes of quantum complexity and phase transitions in the Cluster-XY model.

lines that separate different quantum phases, with enhanced values at the phase boundaries. Notably, the strong correlation between the three quantities suggests that antiflatness and capacity of entanglement serve as reliable indicators of phase transitions, akin to nonstabilizerness measures.

Overall, the phase diagram reveals the intricate competition between the cluster interaction and XY couplings, leading to a rich variety of phases. The results provide a comprehensive characterization of entanglement structure and nonstabilizerness in the Cluster-XY model, further supporting the utility of stabilizer-based diagnostics in identifying quantum critical behavior.

Figure 13 provides a detailed analysis of the SRE $m_2$, the logarithm of the spectrum's antiflatness $\log(\Lambda)$, and the capacity of entanglement $C_E$ as functions of the parameter $\lambda_y$, for different system sizes $L = 16, 32, 64, 128$. The top row focuses on the case where $h = 0$, while the bottom row corresponds to $h = 1$. Across both rows, a common feature emerges: all three quantities exhibit distinct peaks at the critical points, signaling enhanced quantum complexity and nonstabilizerness at these transitions. The peaks sharpen and become more pronounced as the system size increases, consistent with the expected behavior at quantum phase transitions. In the first row, where $h = 0$, the presence of multiple peaks suggests the existence of intricate phase transitions driven by competition between the cluster interaction and the XY couplings. The SRE $m_2$ displays a nontrivial structure, revealing regions of strong nonstabilizerness beyond the cluster phase. The antiflatness highlights the spectral properties of entanglement,

further confirming that the complexity of the wavefunction increases significantly at the phase boundaries. The capacity of entanglement, which serves as a measure of entanglement fluctuations, shows similar peaks, reinforcing the interpretation that these transitions correspond to significant reorganizations of the entanglement structure. In the second row, where $h = 1$, the qualitative behavior remains similar, but the structure of the peaks is somewhat simpler, indicating that the transverse field reduces the number of competing phases. Nevertheless, the scaling with $L$ remains evident, showing that these transitions persist in the thermodynamic limit. These results collectively demonstrate that stabilizer Rényi entropy, antiflatness, and entanglement capacity are powerful diagnostics for identifying and characterizing phase transitions in spin chains, particularly in models where nonstabilizerness plays a fundamental role.

## 5  Conclusion

In this work, we have presented a comprehensive investigation into the quantum complexity of spin-1/2 models, focusing on the interplay between entanglement and nonstabilizerness (magic). By analyzing entanglement spectral quantities (antiflatness and capacity of entanglement) and their relation to magic, we have provided new insights into the rich phase diagrams of various spin models, including the XXZ model, the transverse field XY model (both with and without Dzyaloshinskii-Moriya interactions), and cluster Ising and cluster XY models. Our results complement the rich literature on entanglement and magic in spin chains, that was thus far mostly disconnected.

Combined with the recent results obtained in literature over the last few years, our findings demonstrate that entanglement spectral measures and nonstabilizerness are strongly intertwined, and are effective tools for characterizing the criticality and phase structure of quantum many-body systems. Specifically: (i) entanglement and magic have been shown to independently provide valuable insights into the properties of quantum phases: in most cases under investigations, magic and entanglement capacity show very similar qualitative features, a highly non-trivial fact given that the first is in fact a basis-dependent quantity. However, their interplay reveals deeper features of quantum complexity that are invisible when these resources are considered in isolation. (ii) Antiflatness and capacity of entanglement effectively differentiate between quantum phases, highlight critical points, and identify regions of increased complexity across all studied models.

Looking ahead, our results open the door to further exploration of quantum complexity in more exotic settings, including higher-dimensional systems, open quantum systems, and models with long-range interactions. Furthermore, the application of our methodology to experimentally realizable platforms, such as trapped ions, Rydberg atoms, and superconducting qubits, could provide a bridge between theoretical insights and practical implementations of quantum technologies.

Ultimately, our work highlights the critical role of nonstabilizerness in understanding the emergence of quantum complexity and lays the groundwork for future studies on the interplay of magic, entanglement, and criticality in quantum many-body systems.

## Acknowledgements

We thank T. Chanda, M. Collura, B. Jasser, G. Lami, L. Leone, J. Odavić, S. Oliviero, and P. Tarabunga for discussions and collaborations on a previous work that inspired the present one. E. T. acknowledges collaboration with X. Turkeshi and P. Sierant on related subjects. E. T.

acknowledges support from ERC under grant agreement n.101053159 (RAVE), and CINECA (Consorzio Interuniversitario per il Calcolo Automatico) award, under the ISCRA initiative and Leonardo early access program, for the availability of high-performance computing resources and support. M. D. was partly supported by the QUANTERA DYNAMITE PCI2022-132919, by the EU-Flagship programme Pasquans2, by the PNRR MUR project PE0000023-NQSTI, the PRIN programme (project CoQuS), and the ERC Consolidator grant WaveNets. A.H. was supported by PNRR MUR Project No. PE0000023-NQSTI. A.H. and M.V. acknowledge financial support from PNRR MUR Project No. CN 00000013-ICSC. M.V. acknowledge computational resources from MUR, PON "Ricerca e Innovazione 2014-2020", under Grant No. PIR01-00011 - (I.Bi.S.Co.).

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

# A   Numerical methods for efficient nonstabilizerness estimation

In this appendix, we present two methods that we used for computing nonstabilizerness: the perfect sampling method [60] and the Pauli MPS method [59]. The perfect sampling method aims to iteratively select the most representative Pauli strings in the Pauli-basis expansion of our target quantum state. In contrast, the Pauli-MPS approach directly encodes the full state in the Pauli basis and leverages that representation to compute its nonstabilizerness. Together, these complementary techniques form a unified framework for efficiently probing and quantifying nonstabilizerness for many-body quantum system of large sizes.

## A.1   Pauli Sampling

We begin by considering a system of $N$ qubits in a pure state $\rho$. Its density matrix can be expanded in the Pauli basis as:

$$\rho = d^{-1} \sum_{\sigma \in \mathcal{P}_N} \text{Tr}(\sigma \rho) \sigma. \tag{53}$$

In Sec. 3.3, we introduced the normalized expectation value $\Xi_\sigma(|\rho\rangle)$ (see Eq. 29) as the probability of finding a Pauli string $\sigma \in \mathcal{P}_N$ in the Pauli expansion of $\rho$. To maintain consistency with the notation in [60], we define:

$$\Pi_\rho(\sigma) = \Xi_\sigma(|\rho\rangle) = d^{-1}\text{Tr}^2(\sigma\rho), \tag{54}$$

which represents the *a priori* probability of finding the Pauli string $\sigma$ in the expansion. Using the chain rule, we can express $\Pi_\rho(\sigma)$ in terms of conditional probabilities:

$$\Pi_\rho(\sigma_1\cdots\sigma_N) = \pi_\rho(\sigma_1)\pi_\rho(\sigma_2|\sigma_1)\pi_\rho(\sigma_3|\sigma_1\sigma_2)\cdots\pi_\rho(\sigma_N|\sigma_1\cdots\sigma_{N-1}), \tag{55}$$

where:

$$\pi_\rho(\sigma_1\cdots\sigma_j) = d^{-1} \sum_{\boldsymbol{\sigma} \in \mathcal{P}_{N-j}} \text{Tr}^2(\rho\sigma_1\cdots\sigma_j\boldsymbol{\sigma}) \tag{56}$$

gives the probability of finding the partial Pauli string $\sigma_1\cdots\sigma_j$ in the expansion of $\rho$, and

$$\pi_\rho(\sigma_j|\sigma_1\cdots\sigma_{j-1}) = \frac{\pi_\rho(\sigma_1\cdots\sigma_j)}{\pi_\rho(\sigma_1\cdots\sigma_{j-1})} \tag{57}$$

denotes the conditional probability of finding $\sigma_j$ given that the first $j-1$ terms in the expansion are fixed to the partial Pauli string $\sigma_1\cdots\sigma_{j-1}$.

Conditioning on the event of finding the partial Pauli string $\sigma_1\cdots\sigma_{j-1} \in \mathcal{P}_{N-j+1}$ in the Pauli expansion of $\rho$ corresponds to evaluating $\rho$ in the latter partial Pauli string:

$$\rho \mid_{\sigma_1\cdots\sigma_j} = d^{-1} \sum_{\boldsymbol{\sigma} \in \mathcal{P}_{N-j}} \text{Tr}(\rho\sigma_1\cdots\sigma_j\boldsymbol{\sigma})\sigma_1\cdots\sigma_j\boldsymbol{\sigma}. \tag{58}$$

This allows us to construct the partially projected state:

$$\rho_{j-1} = \frac{\rho \mid_{\sigma_1\cdots\sigma_{j-1}}}{\pi_\rho(\sigma_1\cdots\sigma_{j-1})^{1/2}}. \tag{59}$$

This formulation provides a natural interpretation of the conditional probability in Eq. 57, which can be though as the probability of finding $\sigma_j$ in the Pauli expansion of partially projected state $\rho_{j-1}$. In formulas:

$$\pi_\rho(\sigma_j|\sigma_1\cdots\sigma_{j-1}) = \pi_{\rho_{j-1}}(\sigma_j) \tag{60}$$

We can, then, introduce the following recursive relation:

$$\rho_j = \pi_{\rho_{j-1}}(\sigma_j)\rho_{j-1}\big|_{\sigma_j}. \tag{61}$$

which can be used to iteratively compute the normalized expectation value $\Pi_\rho(\sigma)$.

In many settings, such as for ground states of gapped Hamiltonians and states obeying an entanglement area-law, it is common to express quantum states in the MPS formalism [8]. In the context of magic estimation, this formalism allows for exploiting the local structure of MPS to iteratively sample Pauli strings by computing conditional probabilities (see Eq. 57) and using the chain rule (Eq. 55). To begin, we express the probability distribution over Pauli strings as:

$$\Pi_\rho(\sigma) = d^{-1}\mathrm{Tr}^2(\sigma\,|\rho\rangle\langle\rho|) = d^{-1}\langle\rho|\,\sigma\,|\rho\rangle\,\langle\rho^*|\,\sigma^*|\rho^*\rangle = d^{-1}\mathrm{Tr}(\sigma\otimes\sigma^*\rho\otimes\rho^*), \tag{62}$$

which will be useful in the following derivation. Let us assume that $|\rho\rangle$ is represented as a right-normalized MPS:

$$|\rho\rangle = \sum_{s_1,\dots,s_N} \mathbb{A}_1^{s_1}\mathbb{A}_2^{s_2}\cdots\mathbb{A}_N^{s_N}\,|s_1 s_2\cdots s_N\rangle, \tag{63}$$

where $s_i$ with $i\in\{1,\dots,N\}$ are site indices, and $\mathbb{A}_j^{s_j}$ are $\chi\times\chi$ matrices (except for $\mathbb{A}_1^{s_1}$ and $\mathbb{A}_N^{s_N}$, which are $1\times\chi$ and $\chi\times 1$ matrices, respectively).

To sample the probability distribution $\{\Pi_\rho(\sigma)\}$ efficiently, we rewrite the expectation value $\langle\sigma\rangle_\rho = \mathrm{Tr}(\sigma\rho)$ exploiting the local structure of MPS:

$$\langle\sigma\rangle_\rho = \sum_{s_1,\dots,s_N,s_1',\dots,s_N'} \mathbb{A}_1^{s_1}\mathbb{A}_1^{s_1'*}\langle s_1'|\,\sigma_1\,|s_1\rangle\cdots\mathbb{A}_N^{s_N}\mathbb{A}_N^{s_N'*}\langle s_N'|\,\sigma_N\,|s_N\rangle. \tag{64}$$

For conciseness, we introduce the notation:

$$\langle\sigma\rangle_\rho = \mathbb{T}_1^{(\sigma_1)}\cdots\mathbb{T}_N^{(\sigma_N)}, \tag{65}$$

where the tensors $\mathbb{T}_j^{(\sigma_j)}$ are defined as:

$$\mathbb{T}_j^{(\sigma_j)} = \sum_{s_j,s_j'} \mathbb{A}_j^{s_j}\mathbb{A}_j^{s_j'*}\langle s_j'|\,\sigma_j\,|s_j\rangle. \tag{66}$$

These are local, hypercubic, order-4 tensors of dimension $\chi$ (except $\mathbb{T}_1^{(\sigma_1)}$ and $\mathbb{T}_N^{(\sigma_N)}$, which have dimensions $1\times 1\times\chi\times\chi$).
We now iteratively sample the Pauli string $\sigma$ by computing conditional probabilities. The first step is to determine the probability of a Pauli operator $\sigma_1$ appearing in the Pauli expansion of $\rho$:

$$\pi_\rho(\sigma_1) = d^{-1}\sum_{\boldsymbol{\sigma}\in\mathcal{P}_{N-1}}\mathrm{Tr}^2(\rho\sigma_1\boldsymbol{\sigma}) = \frac{1}{2}\mathrm{Tr}(\sigma_1\otimes\sigma_1^*\rho\otimes\rho^*) = \frac{1}{2}\mathbb{T}_1^{(\sigma_1)}\mathbb{T}_1^{(\sigma_1)*}. \tag{67}$$

We compute the probability distribution $\{\pi_\rho(\sigma_1)\}_{\sigma_1}$, then we extract form this distribution a Pauli operator $\sigma_1$. To account for this partial projection, authors in [60] introduce the operator:

$$\mathbb{L}_1 = \frac{1}{\sqrt{2\pi_\rho(\sigma_1)}}\mathbb{T}_1^{(\sigma_1)}. \tag{68}$$

This allows to compute conditional probabilities at subsequent sites. As an example, the probability of obtaining $\sigma_2$ given that $\sigma_1$ has already been sampled is:

$$\pi_\rho(\sigma_2|\sigma_1) = \frac{1}{2}\mathbb{L}_1\mathbb{T}_2^{(\sigma_2)}\mathbb{T}_2^{(\sigma_2)*}\mathbb{L}_1^*. \tag{69}$$

More generally, for any site $i$, the conditional probability of sampling $\sigma_i$ given the previously sampled Pauli operators is:

$$\pi_\rho(\sigma_i|\sigma_1\cdots\sigma_{i-1}) = \frac{1}{2}\mathbb{L}_{i-1}\mathbb{T}_i^{(\sigma_i)}\mathbb{T}_i^{(\sigma_i)*}\mathbb{L}_{i-1}^*. \tag{70}$$

At each step, the operator $\mathbb{L}$ is updated as follows:

$$\mathbb{L}_i = \frac{1}{\sqrt{2\pi_\rho(\sigma_i|\sigma_1\cdots\sigma_{i-1})}}\mathbb{L}_{i-1}\mathbb{T}_i^{(\sigma_i)}. \tag{71}$$

By iterating this process over all $N$ sites, we efficiently sample a Pauli string $\sigma$. The complexity of sampling a Pauli string has been proven in [60] to be $O(N\chi^3)$.

The estimation error of magic can be make arbitrarily close to the theoretical value by increasing the number $\mathcal{N}$ of samples. However, in practical settings, $M_1$ and $M_2$ have been numerically proven to be estimated with an error of order $10^{-3}$ with order $10^5$ samples.

## A.2   Pauli-MPS

Since the SREs are expressed in terms of Pauli expectation values, it would be advantageous to store the expectation values in an efficient way. This can be achieved simply by representing the state in the Pauli basis. If a state is efficiently represented by an MPS with bond dimenion $\chi$, its exact representation in the Pauli basis can also be written as an MPS with bond dimension $\chi^2$, i.e., the Pauli-MPS [59]. This new approach has been introduced to quantify nonstabilizerness in MPS representations by expressing the state directly in the Pauli basis. In this formalism, the SRE of index $n$ can be evaluated as the contraction of $2n$ replicas of Pauli-MPS. Given an MPS representation of a quantum state $|\psi\rangle$

$$|\psi\rangle = \sum_{s_1,s_2,\cdots,s_N} A_1^{s_1}A_2^{s_2}\cdots A_N^{s_N}|s_1,s_2,\cdots s_N\rangle \tag{72}$$

where $A^{s_i}$ are the local tensors, the key idea is to expand the state in terms of Pauli operators. The Pauli strings $P_\alpha$ form a complete basis for quantum states. The coefficients of this expansion, known as the Pauli spectrum, can be encoded in an MPS structure:

$$|P(\psi)\rangle = B^{\alpha_1}B^{\alpha_2}\cdots B^{\alpha_N}|\alpha_1\alpha_2\cdots\alpha_N\rangle \tag{73}$$

where $B_i^{\alpha_i} = \sum_{s,s'}\langle s|P_{\alpha_i}|s'\rangle A_i^s\otimes\overline{A_i^{s'}}/\sqrt{2}$ are $\chi^2\times\chi^2$ matrices. Note that the MPS is normalized due to the relation $\frac{1}{2^N}\sum_\alpha\langle\psi|P_\alpha|\psi\rangle^2 = 1$ which holds for pure states. Moreover, it retains the right normalization, due to the identity $\frac{1}{2}\sum_\alpha P_\alpha(\cdot)P_\alpha = \mathbb{1}\operatorname{Tr}[\cdot]$. Consequently, the entanglement spectrum of $|P(\psi)\rangle$ is given by $\lambda'_{i,j} = \lambda_i\lambda_j$ for $i,j = 1,2,\cdots,\chi$, where $\lambda_i$ is the entanglement spectrum of $|\psi\rangle$, and hence the von Neumann entropy is doubled.

As we show below, the MPS representation in the Pauli basis provides a powerful and versatile tool to compute the SRE. To do so, we define a diagonal operator $W$ whose diagonal elements are the components of the Pauli vector, $\langle\alpha'|W|\alpha\rangle = \delta_{\alpha',\alpha}\langle\alpha'|P(\psi)\rangle$. The MPO form of $W$ reads

$$W = \sum_{\alpha,\alpha'}\overline{B}_1^{\alpha_1,\alpha'_1}\overline{B}_2^{\alpha_2,\alpha'_2}\cdots\overline{B}_N^{\alpha_N,\alpha'_N}|\alpha_1,\cdots,\alpha_N\rangle\langle\alpha'_1,\cdots,\alpha'_N| \tag{74}$$

where $\overline{B}_i^{\alpha_i,\alpha'_i} = B_i^{\alpha_i}\delta_{\alpha_i,\alpha'_i}$. Applying $W$ $n-1$ times to $|P(\psi)\rangle$, we obtain $|P^{(n)}(\psi)\rangle = W^{n-1}|P(\psi)\rangle$, which is a vector with elements $\langle\alpha|P^{(n)}(\psi)\rangle = \langle\psi|P_\alpha|\psi\rangle^n/\sqrt{2^{Nn}}$. We denote the local tensors of $|P^{(n)}(\psi)\rangle$ by $B_i^{(n)\alpha_i}$. We have

$$\frac{1}{2^{Nn}}\sum_\alpha\langle\psi|P_\alpha|\psi\rangle^{2n} = \langle P^{(n)}(\psi)|P^{(n)}(\psi)\rangle \tag{75}$$

and

$$M_n = \frac{1}{1-n} \log \langle P^{(n)}(\psi) | P^{(n)}(\psi) \rangle - N. \tag{76}$$

Compared to the original replica trick formulation [58], the advantage of replica Pauli-MPS method comes in two aspects. First, the physical dimension of the intermediate MPS to compute the SRE of index $n$ is constantly $d^2$ using the replica Pauli-MPS approach, while Ref. [58] requires a physical dimension of $d^{2(n-1)}$, which grows exponentially with $n$. Since the bond dimension is $\chi^{2n}$ in both methods, the cost of exact contraction to compute the SRE is $O(Nd^2\chi^{6n})$ and $O(Nd^{2(n-1)}\chi^{6n})$, respectively. Second, since the intermediate MPS in the replica Pauli-MPS method is obtained by repeated MPO-MPS multiplication, one can sequentially compress the resulting MPS after every iteration using standard tensor network routines.