# Peer review of "Interplay of entanglement structures and stabilizer entropy in spin models"

_SciPost Physics Core_

## Round 1 · Referee Report · Anonymous (Referee 1) · 2025-7-25

Strengths

  1. The introduction provides an excellent pedagogical overview of the research field, offering comprehensive background material.

  2. The manuscript presents extensive numerical results on nonstabilizerness across various spin-1/2 models, encompassing several systems that have not been previously investigated in the literature.

Weaknesses

  1. While this may be inevitable given the nature of this manuscript, the motivation for examining each model and the contributions obtained from such analysis are not clearly articulated, if any.

Report

In this paper, the authors present a comprehensive study of nonstabilizerness in various spin-1/2 quantum spin chains. The authors numerically compute the stabilizer Rényi entropy (SRE) and the antiflatness of entanglement spectra to demonstrate their effectiveness in distinguishing different phases of matter. Specifically, the authors provide detailed analyses of the XXZ model, the XY model with and without Dzyaloshinskii-Moriya interactions, the cluster Ising model, and the cluster XY model. Overall, I believe the manuscript constitutes an important contribution to the field and merits publication in SciPost Physics Core.

The paper is well-organized and includes a useful pedagogical review of the research area in the introduction and at the beginning of sections. To strengthen the content and improve readability, I request several improvements in the presentation of the manuscript as outlined below.

Requested changes

  1. (lines 372 ~) The character $d$ is defined differently from the previous subsection, where it was originally defined as the dimension of the local Hilbert space. Here, it should represent the dimension of the total Hilbert space, i.e., $d=2^n$ for $n$-qubit systems.

  2. (lines 511~) The explanation of the phase diagram for the XXZ model is somewhat confusing. In the definition of the Hamiltonian (Eq. (32)), the constant $J$ is introduced. However, the authors give the phase transition lines as $h_s=1+\Delta$ and $h_c$, which should actually depend on $J$. I request revision to clarify the discussion in this section. Additionally, if the spin chain is subject to periodic boundary conditions, this should be explicitly stated.

  3. (lines 519~) I do not understand the sentence "In the ferromagnetic phase $\Delta<1$, the SRE is minimal,..." because in Fig. 3(a) the SRE appears to be minimal in the region $0<\Delta$. Perhaps I have misunderstood something, or the horizontal axis in the plots is labeled incorrectly.

  4. (lines 549 ~) The authors assume open boundaries for the XY model, but is the classical-like ground state still obtained for open chains? Ref. [134] did not specify the boundary conditions in the model, but later assumed translational symmetry, which appears to be valid only for periodic chains.

  5. (page 20 Fig. 8) The legends for the plots are illegible. It would be better to insert the equations manually in the plot area. Another minor comment is that the fitting curves should be moved to the background so that the data points appear in the foreground.

Recommendation

Publish (meets expectations and criteria for this Journal)

  • validity: high
  • significance: good
  • originality: good
  • clarity: good
  • formatting: good
  • grammar: good

Author:  Michele Viscardi  on 2025-12-02  [id 6095]

(in reply to Report 1 on 2025-07-25)
Category:
answer to question

We sincerely thank the reviewer for their thoughtful comments and suggestions. We have prepared a detailed document in which each comment is addressed individually. Below, we summarize the main changes made to the manuscript: - line 261: added definition of $d$ and $d_A$; - lines 273-286: improved the clarity of the paragraph; - added footnote 3 and lines 528-529 to specify our bipartition of choice for the spectral quantities mentioned in the manuscript; - lines 622-632: added a clarification about the magic localisation phenomenon in the quantum XY model; - updated the references; - line 383: specified the meaning of $d$; - lines 517-526: clarified parameters of the XXZ Hamiltonian and the characteristics of its phase diagram; - added footnote 4; - removed Figure 8b.

Attachment:

Answer_Reviewer_2.pdf

---

## Round 1 · Referee Report · Anonymous (Referee 2) · 2025-7-30

Strengths

  • The manuscript, in its introduction, presents an excellent review of the research field.

  • The manuscript investigates a broad variety of one dimensional spin models.

Report

This manuscript presents a comprehensive study of nonstabilizerness in various spin-1/2 quantum spin chains. The authors numerically evaluate the stabilizer Rényi entropy (SRE) and the antiflatness of entanglement spectra to demonstrate their effectiveness in distinguishing different quantum phases and the interplay between entanglement structures and SRE. They also provide, in the introduction, an excellent review of the research field and , in the supplementary material, a summary of the state-of-the-art techniques for the numerical evaluation of SREs.

I believe the manuscript represents a significant contribution to the field and I strongly recommend its publication in SciPost Physics Core, after the implementation of the changes requested below.

Requested changes

1- Equation 20 and line 259: what are d and dA? Are they the dimension of the Hilbert spaces of total system and of A? These quantities should be defined. 2 - Lines 268-278: this paragraph was hard to follow. Therefore, I would recommend the authors to revise it for improved clarity and flow. 3 - In Section 4: Which bipartition are the authors considering for the computation of entanglement entropy / antiflatness / capacity of entanglement? I would assume it to be half chain, but it should be specified. 4- In the analysis of the XY chain in Section 4.2 the authors report a change in the behaviour of nonstabilizerness and antiflatness moving from the regime of small anisotropy to larger one. I appreciate their argument explaining the transition from localised to non-localised magic. However, I keep wondering what is the physical aspect that makes this shift possible? At first I believed that the oscillations could be related to the oscillations in the parity of the ground-state happening below the separability line, but then one would have to observe similar oscillations also for larger values of $\gamma$. So I wonder if this is, instead, somehow related to the fact that at $\gamma=0.01$ we are very close to the other transition line of the model, and hence we observe the change in complexity?

Recommendation

Publish (meets expectations and criteria for this Journal)

  • validity: high
  • significance: good
  • originality: good
  • clarity: good
  • formatting: good
  • grammar: good

Author:  Michele Viscardi  on 2025-12-02  [id 6094]

(in reply to Report 2 on 2025-07-30)
Category:
answer to question

We sincerely thank the reviewer for their thoughtful comments and suggestions. We have prepared a detailed document in which each comment is addressed individually. Below, we summarize the main changes made to the manuscript: - line 261: added definition of $d$ and $d_A$; - lines 273-286: improved the clarity of the paragraph; - added footnote 3 and lines 528-529 to specify our bipartition of choice for the spectral quantities mentioned in the manuscript; - lines 622-632: added a clarification about the magic localisation phenomenon in the quantum XY model; - updated the references; - line 383: specified the meaning of $d$; - lines 517-526: clarified parameters of the XXZ Hamiltonians and the characteristics of its phase diagram; - added footnote 4; - removed Figure 8b.

Attachment:

Answer_Reviewer_1.pdf

---

## Round 2 · Referee Report · Anonymous (Referee 1) · 2025-12-11

Report

The authors have adequately addressed all my concerns from the previous version of the manuscript, and I believe it is now ready for publication. In particular, regarding my question 4 on the applicability of the separability circle for open boundary conditions, the authors have provided convincing evidence that the ground state approaches a separable state in the thermodynamic limit, even in OBC where translational symmetry does not hold. I am satisfied with their clarification on this point.

Recommendation

Publish (easily meets expectations and criteria for this Journal; among top 50%)

---

## Round 2 · Referee Report · Anonymous (Referee 2) · 2026-1-6

Report

I have reviewed the revised version of the manuscript and can confirm that the authors have properly addressed all of the concerns I raised in my first report. In its current form, the paper is ready for publication.

Recommendation

Publish (easily meets expectations and criteria for this Journal; among top 50%)

---

## Editorial Decision

voting_in_preparation